# Centrosome guides spatial activation of Rac to control cell polarization and directed cell migration

Hung-Wei Cheng[1],*, Cheng-Te Hsiao[2],*, Yin-Quan Chen[3,4] ⓘ, Chi-Ming Huang[1] ⓘ, Seng-I Chan[1], Arthur Chiou[4], Jean-Cheng Kuo[1,3] ⓘ

**Directed cell migration requires centrosome-mediated cell polarization and dynamical control of focal adhesions (FAs). To examine how FAs cooperate with centrosomes for directed cell migration, we used centrosome-deficient cells and found that loss of centrosomes enhanced the formation of acentrosomal microtubules, which failed to form polarized structures in wound-edge cells. In acentrosomal cells, we detected higher levels of Rac1-guanine nucleotide exchange factor TRIO (Triple Functional Domain Protein) on microtubules and FAs. Acentrosomal microtubules deliver TRIO to FAs for Rac1 regulation. Indeed, centrosome disruption induced excessive Rac1 activation around the cell periphery via TRIO, causing rapid FA turnover, a disorganized actin meshwork, randomly protruding lamellipodia, and loss of cell polarity. This study reveals the importance of centrosomes to balance the assembly of centrosomal and acentrosomal microtubules and to deliver microtubule-associated TRIO proteins to FAs at the cell front for proper spatial activation of Rac1, FA turnover, lamillipodial protrusion, and cell polarization, thereby allowing directed cell migration.**

## Introduction

Cell migration is a critical process in the development and maintenance of multicellular organisms and is involved in many important cell processes, including tissue formation during embryogenesis, wound healing, and various types of immune response (Franz et al, 2002). In many cases, the orchestrated movement of a cell is required to allow migration to a specific location or locations; this is a complex and highly coordinated process driven by various cell-scale dynamic macromolecular ensembles, one of which is the cytoskeleton system. Initially, migrating cells become polarized toward the direction of movement, and this occurs via reorientation of the microtubule-organizing center (MTOC) including the centrosome and the Golgi apparatus (Nobes & Hall, 1999; Etienne-Manneville & Hall, 2001); this results in the assembly of microtubules at the front of the cell and promotion of the dynamic polymerization of actin to extend a membrane protrusion. Subsequently, the protruding membrane adheres to the ECM via the formation of a number of cellular adhesive organelles, namely, the focal adhesions (FAs). FAs are connected to the actin cytoskeleton and transduce contractile force along the bundles of actin filaments (the stress fibers), which acts on the ECM; the result is a maturation process that pulls the cell body forward. Finally, FA disassembly occurs, and this is accompanied by myosin II–mediated contractile forces that pull the trailing edge of the cell away from the ECM (Huttenlocher et al, 1996; Lauffenburger & Horwitz, 1996; Webb et al, 2002; Ridley et al, 2003). The dynamics of the microtubules, the various actin networks, and the FAs need to be orchestrated in a precise spatial and temporal order to bring about directed cell migration (Gupton & Waterman-Storer, 2006). Any errors that occur during the process of cell migration can result in a range of serious consequences, including intellectual disability, vascular disease, tumor formation, and metastasis (Franz et al, 2002).

FAs are the organelles that allow transient ECM attachment at the cell membrane. FAs start to form when their central component, the integrin receptor, is activated by engagement with the ECM. This is subsequently followed by the recruitment of a series of FA-associated proteins that are able to connect to the actin cytoskeleton (Jockusch et al, 1995; Schwartz et al, 1995; Burridge et al, 1988; Hynes, 2002; Zaidel-Bar et al, 2007; Zaidel-Bar & Geiger, 2010). A subset of nascent FAs (new-born FAs) grows and changes protein composition in a process called FA maturation. Mature FAs then either stabilize or begin to disassemble underneath the cell body and at the rear of the cell. Spatial and temporal control of FA turnover brings about dynamic remodeling of the abundance of various groups of proteins in FAs (Kuo et al, 2011), and these changes allow cells to respond to extracellular cues and to transduce specific signals that bring about the modulation of the cytoskeletal system needed for coordinated and productive cell movement (Wu et al, 2015; Yu et al, 2015). The molecular

[1]Institute of Biochemistry and Molecular Biology, National Yang-Ming University, Taipei, Taiwan    [2]Institute of Biological Chemistry, Academia Sinica, Taipei, Taiwan    [3]Cancer Progression Research Center, National Yang-Ming University, Taipei, Taiwan    [4]Institute of Biophotonics, National Yang-Ming University, Taipei, Taiwan

Correspondence: jckuo@ym.edu.tw
*Hung-Wei Cheng and Cheng-Te Hsiao contributed equally to this work

mechanisms underlying the regulation of FA turnover are complex, and dynamic microtubules are starting to reveal its importance (Stehbens & Wittmann, 2012). It has been reported that microtubules that are able to stochastically undergo switching between phases of growth and phases of shortening probably influence FA turnover by locally modulating Rho GTPase signaling (Waterman-Storer et al, 1999; Ezratty et al, 2005; Chang et al, 2008; Rooney et al, 2010) and clathrin-dependent (Ezratty et al, 2005, 2009; Nishimura & Kaibuchi, 2007; Chao & Kunz, 2009) and clathrin-independent endocytosis (Echarri & Del Pozo, 2006). Thus, microtubules are central to spatially controlled FA dynamics and this, in turn, is involved in modulating the outcome of downstream integrin engagement and eventually control of cell migration.

Microtubules are dynamically orientated by the centrosome and other MTOCs, and this results in centrosomal and acentrosomal networks, respectively. The molecular mechanisms by which centrosomal and acentrosomal microtubules regulate directional migration remain unknown up to present, although the centrosome has long been recognized as defining the front of a cell (Wakida et al, 2010; Zhang & Wang, 2017). To address the above question, we have used three types of human retinal pigment epithelial (RPE) cells, control cells (RPEp53$^{-/-}$ cells [Bazzi & Anderson, 2014]) and acentriolar cells (RPEp53$^{-/-}$SAS6$^{-/-}$ cells [Wang et al, 2015] and RPEp53$^{-/-}$STIL$^{-/-}$ cells [Chen et al, 2017]) to characterize the compositional changes in FAs that occur in response to centrosome disruption. We found that centrosome disruption enhances the assembly of acentrosomal microtubules and these are associated with and involved in the delivery of the Rac1 guanine nucleotide exchange factor (GEF), TRIO (Triple Functional Domain Protein), to FAs; this then promotes excessive Rac1 activation and brings about changes in lamellipodial protrusion, which results in differences in FA turnover. The present study addresses the roles of the centrosome in restricting the distribution of acentrosomal microtubules, which in turn affects the activation of Rac1 at the front of a migrating cell.

# Results

## Organization by the centrosome mediates cell polarization and guides directed cell migration

Previous studies have demonstrated that the cellular position of the centrosome is indicative of cell polarization during directed cell migration (Wakida et al, 2010; Zhang & Wang, 2017). To investigate if assembly of the centrosome mediates cell polarization and guides directed cell migration, we examined the effect in retinal pigment epithelial (RPE) cells of disrupting centriole formation (centrosome assembly). Acentriolar cells are not viable in the presence of the p53 gene (Bazzi & Anderson, 2014); therefore, RPEp53$^{-/-}$ cells, which are null for the p53 gene, were used for this part of the study. SAS6 and STIL are the main components in daughter centriole involved in centriole duplication (Leidel et al, 2005; David et al, 2014; Kim et al, 2014; Arquint & Nigg, 2016); thus, the acentriolar cell lines (RPEp53$^{-/-}$SAS6$^{-/-}$ [Wang et al, 2015] and RPEp53$^{-/-}$STIL$^{-/-}$ cells [Chen et al, 2017]) were generated by knocking out SAS6 and STIL separately, using CRISPR/Cas9 technology. Immunoblotting of total cell lysates validated that SAS6 and STIL had been knocked out in the RPEp53$^{-/-}$SAS6$^{-/-}$ and RPEp53$^{-/-}$STIL$^{-/-}$ cells, respectively (Fig 1A). After knockout, the expression of γ-tubulin, a main component of the pericentriolar material that is involved in mediating microtubule nucleation, was found to still have similar levels in RPEp53$^{-/-}$, RPEp53$^{-/-}$SAS6$^{-/-}$, and RPEp53$^{-/-}$STIL$^{-/-}$ cells (Fig 1A). Immunofluorescence microscopy analysis revealed that there was accumulation of γ-tubulin at the centrosome in RPEp53$^{-/-}$ cells, but this was not true for RPEp53$^{-/-}$SAS6$^{-/-}$ and RPEp53$^{-/-}$STIL$^{-/-}$ cells (Fig 1B). Therefore, knockout of either SAS6 or STIL only disrupts the assembly of centrosome but does not seen to affect the expression of pericentriolar material components.

We next determined if an intact centrosome structure is able to modulate directed cell migration; this was assessed by wound-healing migration assay. We measured the percentage of wound closure over 705 min of migration period and found that cells that were null for SAS6 or STIL showed a marked acceleration in wound closure (Fig 1C). We further tracked the trajectory of single individual cells at wound edge over the same period by tracing the center of their nuclei and found that RPEp53$^{-/-}$SAS6$^{-/-}$ and RPEp53$^{-/-}$STIL$^{-/-}$ cells had a much faster migration speed, but poor directional persistence, compared with the control cells (RPEp53$^{-/-}$) (Fig 1D). These effects could be visualized via displays of representative cell migration paths using window plots; these showed that RPEp53$^{-/-}$SAS6$^{-/-}$ and RPEp53$^{-/-}$STIL$^{-/-}$ cells had longer migration path and changed direction much more frequently than the control cells (Fig 1E). The increased migration path and the reduced directional persistence suggest that the loss of an intact centrosome enhances cell motility but renders these cells unable to maintain a stable direction during targeted migration. The polarization of wound-edge cells, when judged by the reorientation of their Golgi apparatus in the direction of wound, revealed that a significantly lower percentage of RPEp53$^{-/-}$SAS6$^{-/-}$ and RPEp53$^{-/-}$STIL$^{-/-}$ cells in the front row of cells had a polarized Golgi apparatus (Fig 1F). Closer examination of the cells at the wound edge revealed that RPEp53$^{-/-}$ cells displayed a polarized phenotype that included microtubules elongation to the tip of protrusion, whereas on the other hand, RPEp53$^{-/-}$SAS6$^{-/-}$ and RPEp53$^{-/-}$STIL$^{-/-}$ cells showed a scattered microtubule distribution and a decreased percentage of cells with polarized microtubules (Fig 1G). These observations indicate that an intact centrosome does regulate wound-healing migration and it does this by controlling cell polarization and cell motility during directed cell migration.

## Loss of the centrosome enhances the assembly of acentrosomal microtubules

To understand how an intact centrosome controls directed cell migration, we focused on microtubules. We first stained various cells with GOLPH2 (Golgi complex marker) and α-tubulin (a microtubules marker) to visualize the changes in the association of microtubules with the Golgi complex in RPEp53$^{-/-}$SAS6$^{-/-}$ and RPEp53$^{-/-}$STIL$^{-/-}$ cells compared with control cells (Fig 2A). The results support the notion that the Golgi apparatus also serves as an MTOC and is involved in acentrosomal microtubule nucleation and stabilization (Chabin-Brion et al, 2001; Efimov et al, 2007; Zhu & Kaverina, 2013). We have earlier shown that RPEp53$^{-/-}$, RPEp53$^{-/-}$SAS6$^{-/-}$,

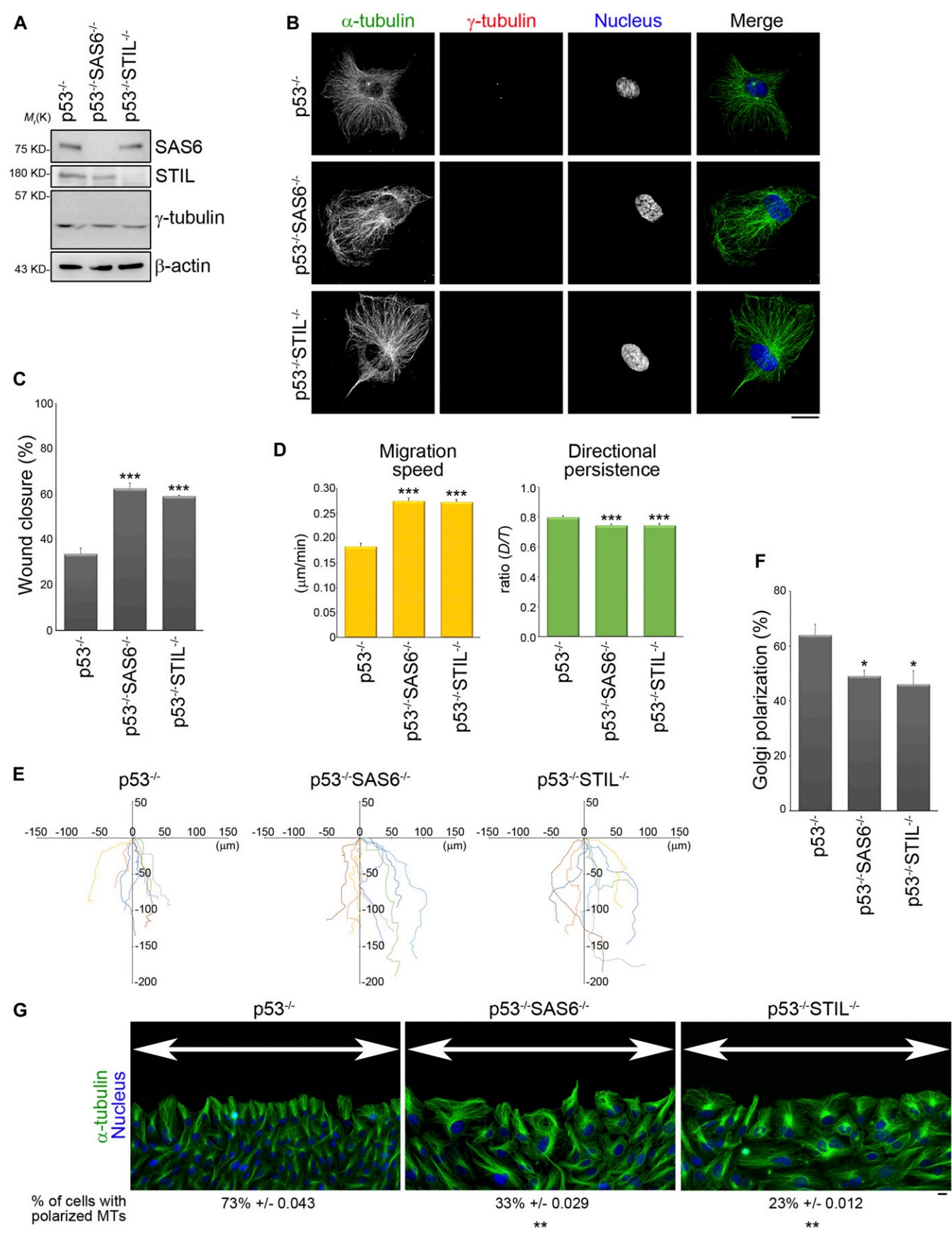

and RPEp53$^{-/-}$STIL$^{-/-}$ cells expressed similar amount of $\alpha$-tubulin in their total cell lysates (Fig 2B) and, therefore, we were able to further characterize whether the centrosome modulated the total amount of polymerized microtubules present in these cells. The polymerized microtubule-containing fraction (pellets; insoluble fraction) and the free tubulin fraction (supernatant; soluble fraction) were isolated from control cells (RPEp53$^{-/-}$ cells) and acentrosomal cells (RPEp53$^{-/-}$SAS6$^{-/-}$ and RPEp53$^{-/-}$STIL$^{-/-}$ cells) using a previously published microtubule isolation method (Sloboda, 2015; see the Materials and Methods section). An actin-binding protein, non-muscle myosin IIA (NMIIA), was mainly detected in soluble fraction by Western blotting (Fig 2B), which confirmed that this microtubule isolation method was able to separate non–microtubule-binding proteins from the polymerized microtubule-containing fraction (insoluble fraction). We compared the abundance of $\alpha$-tubulin in polymerized microtubule-containing fraction (insoluble) and the free tubulin fraction (soluble) by Western blotting, and this revealed that acentrosomal cells contained a higher level of polymerized microtubules and that these were highly acetylated (Fig 2B). Tubulin acetylation has been demonstrated to enhance microtubule stabilization (Portran et al, 2017) and, thus, these findings indicate that loss of the centrosome increases microtubule polymerization and stabilization.

It is known that microtubule plus end dynamics influences cell migration (Waterman-Storer et al, 1999; Wittmann et al, 2003), and, therefore, we further characterized the assembly and disassembly of microtubule plus ends in control cells (RPEp53$^{-/-}$ cells) and in acentrosomal cells (RPEp53$^{-/-}$SAS6$^{-/-}$ and RPEp53$^{-/-}$STIL$^{-/-}$ cells). We performed live-cell imaging using fluorescently labeled EB3 (GFP-EB3), which is known to label the growing plus tips of microtubules (Fig 2C and Video 1–3). In the time-lapse image series showing GFP-EB3, we tracked the dynamics of the growing plus tips of the microtubules using a previously developed software package, namely, plusTip-Tracker (Matov et al, 2010). This allowed us to measure microtubule growth speed and growth persistence (growth lifetime; 1/catastrophe frequency). We then set the median values of the microtubule growth speed and the growth lifetime obtained from the pooled microtubules tracks of RPEp53$^{-/-}$ cells as two thresholds. Based on these thresholds, the pools of total microtubule tracks in RPEp53$^{-/-}$, RPEp53$^{-/-}$SAS6$^{-/-}$, and RPEp53$^{-/-}$STIL$^{-/-}$ cells were classified into two pairs of subgroups, namely, "slow (red)" versus "fast (blue)" and "short-lived (yellow)" versus "long-lived (green)." In the two types of acentrosomal cells, the population of growing microtubules with slow and short-lived

growth was found to be increased, which resulted in an overall trend of microtubule growth in these cells when an intact centrosome was absent (Fig 2D). The loss of a cell's centrosome, thus, significantly decreased mean microtubule growth speed and mean microtubule lifetime compared with control cells (Fig 2E). Thus, the loss of an intact centrosome causes modulation of microtubule plus end dynamics, which can be observed as slowing of microtubule assembly and a reduction in microtubule persistence. These effects can be interpreted as promotion of microtubule dynamic instability.

To characterize actin network remodeling that occurs in acentrosomal cells, the cells were immunostained to detect both F-actin (phalloidin staining) and microtubules ($\alpha$-tubulin staining) using confocal imaging. We observed that in the control (RPEp53$^{-/-}$) cells, there was a dense F-actin meshwork enriched on one side of the cell's periphery that led to a front/back asymmetry distribution associated with membrane protrusions. In contrast, the acentrosomal cells had an enriched polymerized actin meshwork around the cell periphery that gave rise to multiple huge membrane protrusions (Fig 2F). We further examined the role of the centrosome in lamellipodial formation. Immunolocalization in the cells of a lamellipodia marker cortactin showed that loss of the centrosome resulted in a significant increase in the ratio of lamellipodial area to total cell area (Fig 2G). This agrees with a previous study indicating that growing microtubules govern actin network remodeling (Henty-Ridilla et al, 2016), and these findings further suggest that acentrosomal microtubules are able to enhance actin polymerization and the assembly of multiple membrane protrusions. Thus, the centrosome is able to control migrating cell polarization in a microtubule-dependent manner that involves the regulation of the distribution of front/back asymmetry with respect to membrane protrusions, and this occurs via local modulation of the formation of dense actin networks.

### Loss of the centrosome enhances cell adhesion

We next investigated whether assembly of the centrosome controls cell adhesion strength. To do this, we initially examined the adhesive abilities of control (RPEp53$^{-/-}$ cells) and both types of acentrosomal cells (RPEp53$^{-/-}$SAS6$^{-/-}$ and RPEp53$^{-/-}$STIL$^{-/-}$ cells). The cells were seeded on plates coated with fibronectin for 15 min to allow analysis of the cell spreading area (Fig 3A). The results revealed a large spreading area when acentrosomal cells were

Figure 1.  The absence of the centrosome suppresses the cell polarity of a migrating cell.
**(A)** Western blot analysis of cell lysates obtained from RPEp53$^{-/-}$, RPEp53$^{-/-}$SAS6$^{-/-}$, and RPEp53$^{-/-}$STIL$^{-/-}$cells (loaded with equal amounts of total protein) using SAS6, STIL, $\gamma$-tubulin, and $\beta$-actin antibodies. **(B)** Confocal images of immunolocalized $\alpha$-tubulin (to visualize microtubules; green), $\gamma$-tubulin (to visualize centrosome; red), and DAPI (to visualize nucleus; blue) in RPEp53$^{-/-}$, RPEp53$^{-/-}$SAS6$^{-/-}$, and RPEp53$^{-/-}$STIL$^{-/-}$cells. Scale bar, 20 $\mu$m. **(C)** The percentage of wound closure. Data are mean ± SEM (RPEp53$^{-/-}$, n = 4 independent experiments; RPEp53$^{-/-}$SAS6$^{-/-}$, n = 4 independent experiments; and RPEp53$^{-/-}$STIL$^{-/-}$, n = 3 independent experiments). ***$P$ < 0.001, compared with RPEp53$^{-/-}$. **(D)** The migration parameters were calculated as described in the Materials and Methods section. Migration speed was calculated as the total length of the migration path divided by the duration of migration; directional persistence was calculated as the net migration distance divided by the total length of the migration path. Data are mean ± SEM (n = 100 cells for each conditions). ***$P$ < 0.001, compared with RPEp53$^{-/-}$. **(E)** Analysis of migration trajectories. The trajectories of representative cells are plotted. The origins of migration are superimposed at (0, 0). **(F)** The percentage of Golgi reorientation. The percentage of wound-edge cells with their Golgi apparatus in the forward-facing 120° sector was measured at 3 h after wounding. For each experiment, 100 cells were scored. Data are mean ± SEM (n = 3 independent experiments). *$P$ < 0.05, compared with RPEp53$^{-/-}$. **(G)** RPEp53$^{-/-}$, RPEp53$^{-/-}$SAS6$^{-/-}$, and RPEp53$^{-/-}$STIL$^{-/-}$cells were immunostained with $\alpha$-tubulin (green) and DAPI (blue) at 3 h after wounding. The high magnification wide field-of-view (epi-fluorescence) images were generated by automatically stitching multiple adjacent frames from a multipoint using the Nikon analysis software NIS-Elements. Arrows indicate the direction of the wound. Scale bar, 20 $\mu$m. The values indicate the percentage of cells with polarized microtubules. For each experiment, 100 cells were scored. Data are mean ± SEM (n = 3 independent experiments). **$P$ < 0.01, compared with RPEp53$^{-/-}$.

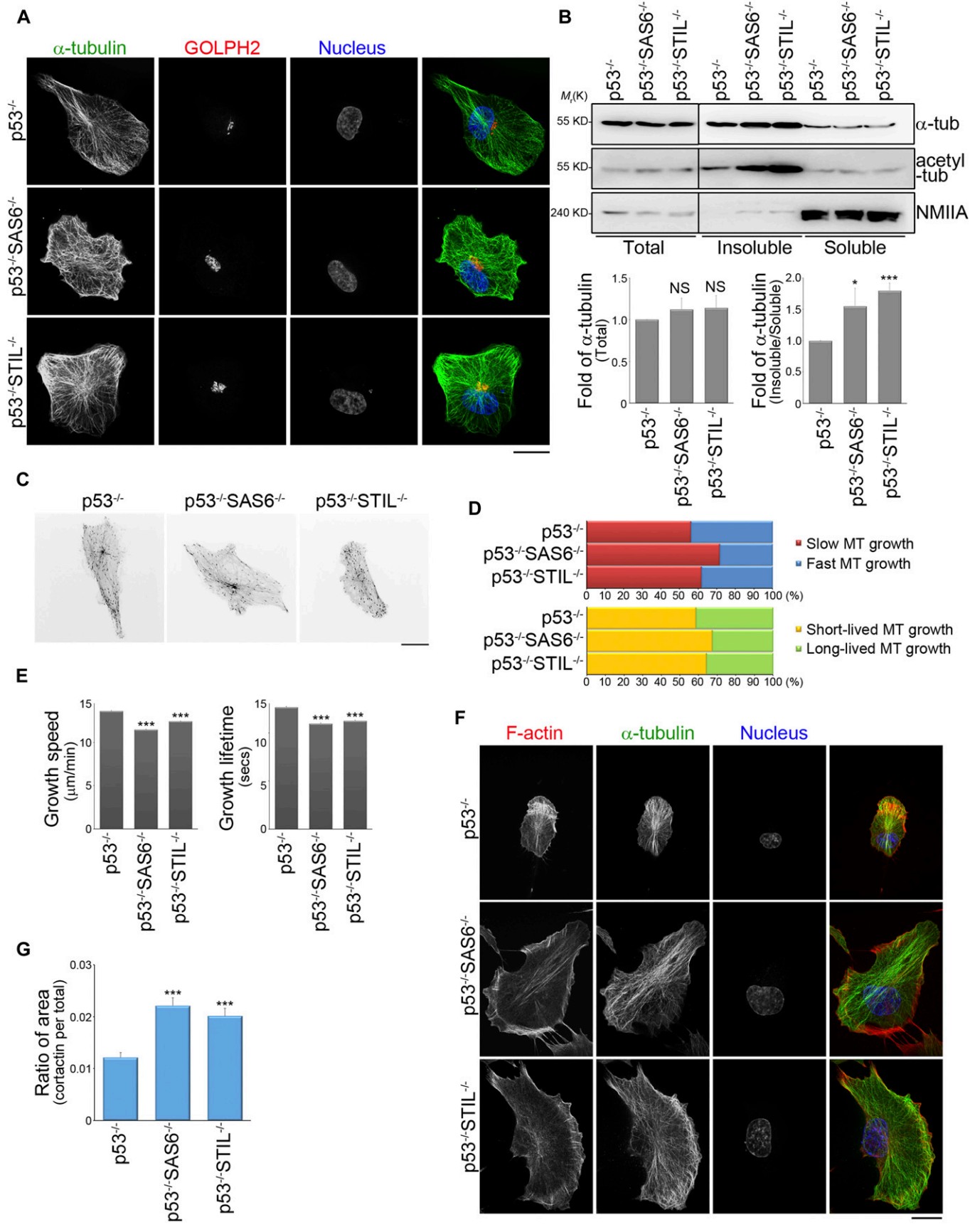

used compared with a smaller spreading area for the control cells (Fig 3B). Next, we further quantified cell adhesiveness and found that the loss of the centrosome increased cell adherence to fibronectin (Fig 3C). Thus, we were able to hypothesize that the microtubules in acentrosomal cells are involved in regulating FA formation and/or FA dynamics to enhance cell adhesion.

To investigate whether microtubules in acentrosomal cells contribute to FA dynamics in a different way to what occurs in control cells, we analyzed the size of FAs in the control and acentrosomal cells using kinetic studies of their recovery after treatment with nocodazole. The cells were incubated in a growth medium containing 10 $\mu$M nocodazole overnight to stop microtubules growth and then fixed at the specific time points after perfusion with nocodazole-free medium that would induce microtubule growth. When untreated, the control condition, we found that acentrosomal cells formed more FAs (RPEp53$^{-/-}$ cells: ~82 FAs/cell, RPEp53$^{-/-}$SAS6$^{-/-}$: ~111 FAs/cell, and RPEp53$^{-/-}$STIL$^{-/-}$ cells: ~135 FAs/cell), which supports the results in Fig 3C and indicates that cells lacking a centrosome are more adhesive. After nocodazole treatment (0 min washout), the cells were found to contain a larger number of big (>2 $\mu$m$^2$) and mid-sized (1–2 $\mu$m$^2$) FAs and a reduced number of small (<1 $\mu$m$^2$) FAs compared with the untreated (control) cells (Fig 3D). At 3 min after perfusion with nocodazole-free medium, we found that the acentrosomal cells showed a significant increase in the number of small FAs and a decrease in the number of big FAs compared to the RPEp53$^{-/-}$ cells (Fig 3D). This suggests that loss of the centrosome promotes FA formation in a microtubule-dependent manner. Analysis of GFP-paxillin dynamics revealed that loss of the centrosome also significantly shortened FA lifetime (Fig 3E and F), especially the time needed for FA assembly and disassembly (Fig 3G). FA turnover has been shown to be regulated by dynamic microtubules attaching to FAs (Stehbens & Wittmann, 2012); thus, we hypothesize that the enhancement in acentrosomal microtubules that is induced by centrosome disruption may deliver signals to FAs and control FA dynamics. Indeed, immunolocalization of paxillin and α-tubulin revealed that FAs were co-localized with microtubules in both the control (RPEp53$^{-/-}$ cells) and acentrosomal cells (RPEp53$^{-/-}$SAS6$^{-/-}$ and RPEp53$^{-/-}$STIL$^{-/-}$ cells) (Fig 3H). Therefore, loss of the centrosome causes promotion of FA formation and increase in the turnover of FAs; this occurs most likely via specific signals delivered from the acentrosomal microtubules.

## Centrosome reduces lamellipodial protrusion factors in FAs

To further characterize how FA-mediated signals are modulated by the centrosome, we analyzed the composition and abundance of proteins in FAs fractionated from control cells (RPEp53$^{-/-}$ cells) and acentrosomal cells (RPEp53$^{-/-}$SAS6$^{-/-}$ and RPEp53$^{-/-}$STIL$^{-/-}$ cells). The cells were plated on fibronectin-coated plates and hypotonically shocked to separate the FA fractions using a previously described FA isolation method (Kuo et al, 2011; Kuo et al, 2012; see the Materials and Methods section); this method had been previously confirmed to preserve the native organization of the FAs. The isolated FA fractions were subject to liquid chromatography (LC)–tandem mass spectrometry (MS/MS) analysis. Following the LC-MS/MS analysis, the results were processed using a software program, *Mascot*, to identify the proteins present. The proteins that were identified in at least two of four replicate runs (four independent experimental runs for each condition) were included in the lists of reproducible proteins, namely, the reproducible lists. These lists indicated that there were 1,235 proteins in the FAs of RPEp53$^{-/-}$ cells (Table S1), 1,233 proteins in the FAs of RPEp53$^{-/-}$SAS6$^{-/-}$ cells (Table S2), and 1,228 proteins in the FAs of RPEp53$^{-/-}$STIL$^{-/-}$ cells (Table S3).

To characterize the effect of the centrosome on the changes in abundance of FA proteins, we evaluated the relative levels of the individual FA protein isolated from the control cells (RPEp53$^{-/-}$ cells) and acentrosomal cells (RPEp53$^{-/-}$SAS6$^{-/-}$ and RPEp53$^{-/-}$STIL$^{-/-}$ cells) using the protein abundance quantification program *Progenesis QI for proteomics*. Variation in experimental conditions meant that the raw abundances (the sums of all unique normalized peptide ion abundances) of each protein on each run quantified by *Progenesis QI for proteomics* were normalized before subsequent analysis (Fig S1). On analysis of the normalized protein abundances of each FA protein between the control cells (RPEp53$^{-/-}$ cells) and the acentrosomal cells (RPEp53$^{-/-}$SAS6$^{-/-}$ and RPEp53$^{-/-}$STIL$^{-/-}$ cells) using the bioinformatics tool, the Ingenuity Pathway Analysis, we uncovered significant changes in the canonical pathways in FAs and that these are highly related to centrosome formation (Fig 4A). These findings indicate that centrosome formation dramatically affects the amount of various proteins present in FAs that are known to be involved in specific pathways; these are the actin cytoskeleton, integrin-based FA, and Rho-family small GTPases (Fig 4A). Finally, the normalized protein

**Figure 2. The absence of the centrosome changes global microtubule dynamics in a migrating cell.**
**(A)** Confocal images of immunolocalized α-tubulin (green), GOLPH2 (to visualize Golgi; red), and DAPI (blue) in RPEp53$^{-/-}$, RPEp53$^{-/-}$SAS6$^{-/-}$, and RPEp53$^{-/-}$STIL$^{-/-}$ cells. Scale bar, 20 $\mu$m. **(B)** Top: pellets (polymerized microtubule-containing fractions; insoluble) and supernatants (free tubulin fractions; soluble) from RPEp53$^{-/-}$, RPEp53$^{-/-}$SAS6$^{-/-}$, and RPEp53$^{-/-}$STIL$^{-/-}$ cells were fractionated and then analyzed by Western blotting using antibodies against α-tubulin (α-tub), acetylated tubulin (acetyl-tub), and NMIIA. Total microtubules (total cell lysate; total) from RPEp53$^{-/-}$, RPEp53$^{-/-}$SAS6$^{-/-}$, and RPEp53$^{-/-}$STIL$^{-/-}$ cells (loaded with equal amount of total protein) are shown. Bottom: (left) fold enrichment of α-tubulin in the total cell lysate determined by Western blotting (n = 5 independent experiments); (right) fold of α-tubulin in the polymerized microtubule-containing fractions (pellet; insoluble): free tubulin fractions (supernatant; soluble) as determined by Western blotting (n = 7 independent experiments). Data are mean ± SEM. *$P$ < 0.05; ***$P$ < 0.001; NS, no significance, all compared with RPEp53$^{-/-}$. **(C)** Confocal images of fluorescent EB3 comets obtained from 2-min time-lapse movies of GFP-EB3 (frame rate = 2 s) from representative cells to allow comparison between RPEp53$^{-/-}$, RPEp53$^{-/-}$SAS6$^{-/-}$, and RPEp53$^{-/-}$STIL$^{-/-}$ cells. Scale bar, 20 $\mu$m. **(D)** Percentage of the population of microtubules whose dynamics were categorized into subpopulations based on the median values for microtubule growth speed and growth excursion lifetime in RPEp53$^{-/-}$ cells. **(E)** Comparison of the average values for microtubule growth speed and growth excursion lifetime obtained from RPEp53$^{-/-}$, RPEp53$^{-/-}$SAS6$^{-/-}$, and RPEp53$^{-/-}$STIL$^{-/-}$ cells. Data are mean ± SEM (RPEp53$^{-/-}$, n = 24,339 comets/10 cells; RPEp53$^{-/-}$SAS6$^{-/-}$, n = 37,006 comets/11 cells; and RPEp53$^{-/-}$STIL$^{-/-}$, n = 43,213 comets/14 cells). ***$P$ < 0.001, all compared with RPEp53$^{-/-}$. **(F)** Confocal images of immunolocalized phalloidin (red), α-tubulin (green), and DAPI (blue) in RPEp53$^{-/-}$, RPEp53$^{-/-}$SAS6$^{-/-}$, and RPEp53$^{-/-}$STIL$^{-/-}$ cells. Scale bar, 20 $\mu$m. **(G)** Ratio of cortactin-stained area (lamellipodia area) relative to total cell area in RPEp53$^{-/-}$, RPEp53$^{-/-}$SAS6$^{-/-}$, and RPEp53$^{-/-}$STIL$^{-/-}$ cells. Data are mean ± SEM (RPEp53$^{-/-}$, n = 29 cells; RPEp53$^{-/-}$SAS6$^{-/-}$, n = 31 cells; and RPEp53$^{-/-}$STIL$^{-/-}$, n = 24 cells). ***$P$ < 0.001, compared with RPEp53$^{-/-}$.

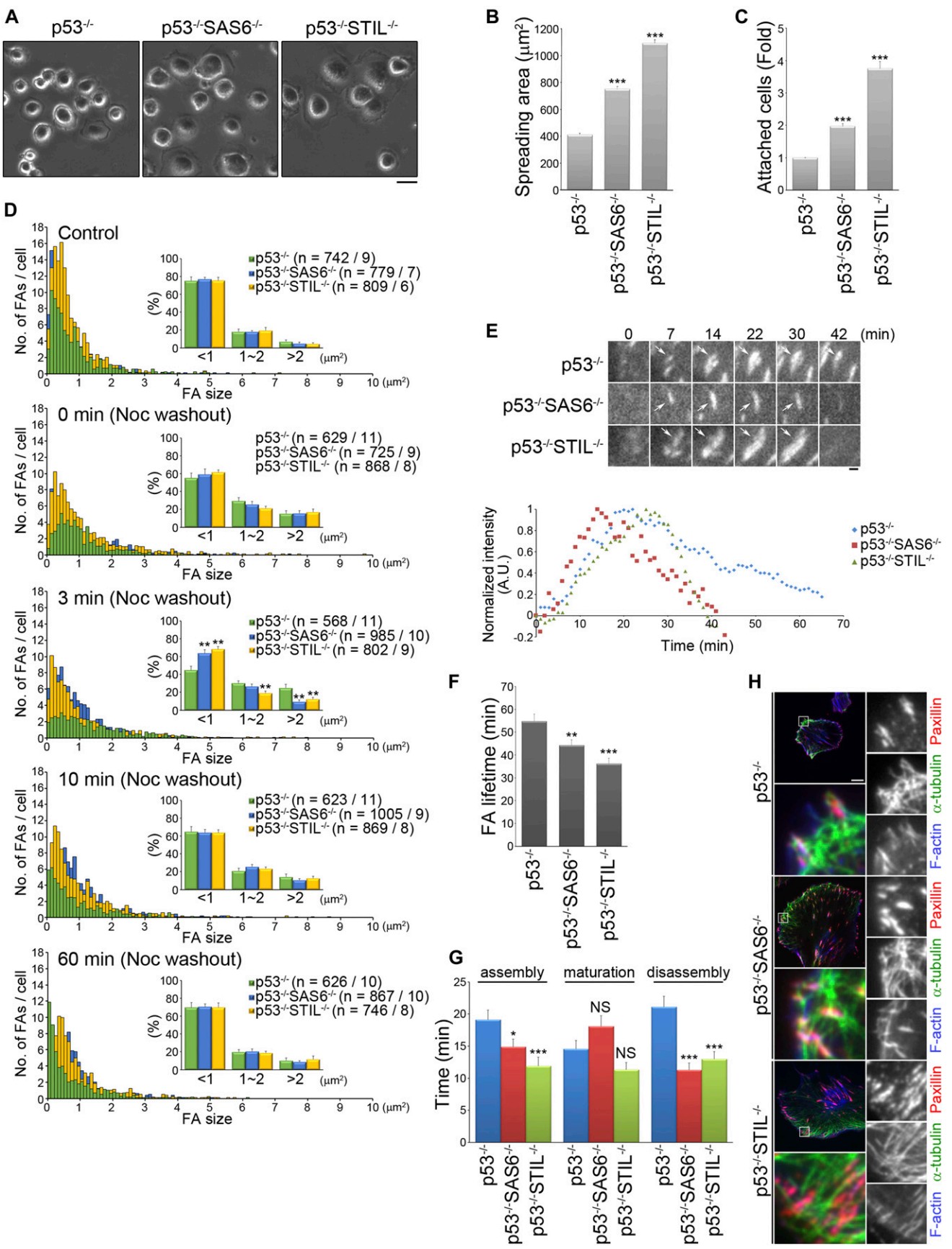

abundance of each protein on each run was calculated to determine the differences (*P*-value; *t* test) in abundance of each protein between the FAs of control cells and the acentrosomal cells (RPEp53$^{-/-}$ versus RPEp53$^{-/-}$SAS6$^{-/-}$; RPEp53$^{-/-}$ versus RPEp53$^{-/-}$STIL$^{-/-}$); these relative protein abundances were calculated as a ratio (the centrosome dependence ratio) to display the centrosome-modulated FA proteome (Tables S4 and S5). Of the proteins reproducibly identified in FAs of the control cells and the acentrosomal cells, those showing significant differences (*P*-value < 0.05) in abundance between the control and acentrosomal cells (RPEp53$^{-/-}$ versus RPEp53$^{-/-}$SAS6$^{-/-}$ or RPEp53$^{-/-}$ versus RPEp53$^{-/-}$STIL$^{-/-}$) were proteins that had ratio >1, indicating centrosome-dependent FA recruitment, and proteins that had ratio <1, indicating FA recruitment was inhibited by centrosome formation. The proteins without any significant difference (*P*-value ≥ 0.05) are believed to represent minor centrosome-dependent changes in FA abundance and centrosome-independent FA recruitment. Thus, centrosome formation was shown to be able to influence FA enrichment of proteins, and this then leads to substantial changes in FA composition. This in turn modulates distinct pathways, namely, microtubule dynamics, integrin-based FA signaling, actin organization, and directed cell migration.

Directed cell migration in living cells is known to be regulated dynamically by Rho-family small GTPases. In the centrosome-modulated FA proteome (Tables S4 and S5), we found that proteins known to be involved in pathways related to Rac1-mediated actin polymerization and microtubule dynamics were strongly enriched in FAs of acentrosomal cells, compared with control cells (Fig 4B). Loss of the centrosome resulted in enrichment in FAs of Rac1, of Rac1 enhancers, of Rac1 effectors, and of various Rac1 downstream targets. The Rac1 enhancers that were recruited to the FAs of acentrosomal cells included Rac GEF TRIO (van Rijssel et al, 2012) (RPEp53$^{-/-}$ versus RPEp53$^{-/-}$SAS6$^{-/-}$: ratio = 0.6224, *P*-value = 0.0097; RPEp53$^{-/-}$ versus RPEp53$^{-/-}$STIL$^{-/-}$: ratio = 0.4299, *P*-value = 0.0131) and Rac GEF modulator EPS8 (epidermal growth factor receptor kinase substrate 8) (Innocenti et al, 2002) (RPEp53$^{-/-}$ versus RPEp53$^{-/-}$SAS6$^{-/-}$: ratio = 0.3979, *P*-value = 0.0061; RPEp53$^{-/-}$ versus RPEp53$^{-/-}$STIL$^{-/-}$: ratio = 0.0977, *P*-value = 0.0021). RhoA inhibitor RhoGDI (Rho GDP-dissociation inhibitor 1) (Del Pozo et al, 2002) (RPEp53$^{-/-}$ versus RPEp53$^{-/-}$SAS6$^{-/-}$: ratio = 0.3178, *P*-value = 0.0224; RPEp53$^{-/-}$ versus RPEp53$^{-/-}$STIL$^{-/-}$: ratio = 0.1336, *P*-value = 0.0022), which indirectly promotes Rac1 activation, was also found to be enriched in FAs when there was centrosome disruption. Furthermore, ARHGAP22 (Sanz-Moreno et al, 2008) (RPEp53$^{-/-}$ versus

RPEp53$^{-/-}$SAS6$^{-/-}$: ratio = 1.3393, *P*-value = 0.0401; RPEp53$^{-/-}$ versus RPEp53$^{-/-}$STIL$^{-/-}$: ratio = 0.304, *P*-value = 0.1944), which inhibits Rac1 activity, was found to be enriched in the FAs of control cells compared with acentrosomal cells. The Rac1 effectors found to be affected included IRSp53 (insulin receptor tyrosine kinase substrate p53) (Miki et al, 2000) (RPEp53$^{-/-}$ versus RPEp53$^{-/-}$SAS6$^{-/-}$: ratio = 0.0975, *P*-value = 0.0246; RPEp53$^{-/-}$ versus RPEp53$^{-/-}$STIL$^{-/-}$: ratio = 0.0217, *P*-value = 0.0154), which serves as a mediator of Arp2/3-dependent actin polymerization. The Rac1 downstream targets involved in actin treadmilling found to be affected included ARP2/3 complex (ARP2 [RPEp53$^{-/-}$ versus RPEp53$^{-/-}$SAS6$^{-/-}$: ratio = 0.6274, *P*-value = 0.1191; RPEp53$^{-/-}$ versus RPEp53$^{-/-}$STIL$^{-/-}$: ratio = 0.182, *P*-value = 0.0173], ARP3 [RPEp53$^{-/-}$ versus RPEp53$^{-/-}$SAS6$^{-/-}$: ratio = 1.0736, *P*-value = 0.3131; RPEp53$^{-/-}$ versus RPEp53$^{-/-}$STIL$^{-/-}$: ratio = 0.4301, *P*-value = 0.0021], ARPC2 [RPEp53$^{-/-}$ versus RPEp53$^{-/-}$SAS6$^{-/-}$: ratio = 1.0516, *P*-value = 0.4124; RPEp53$^{-/-}$ versus RPEp53$^{-/-}$STIL$^{-/-}$: ratio = 0.6521, *P*-value = 0.0292], ARPC1A [RPEp53$^{-/-}$ versus RPEp53$^{-/-}$SAS6$^{-/-}$: ratio = 0.9594, *P*-value = 0.4299; RPEp53$^{-/-}$ versus RPEp53$^{-/-}$STIL$^{-/-}$: ratio = 0.3903, *P*-value = 0.00001], ARPC1B [RPEp53$^{-/-}$ versus RPEp53$^{-/-}$SAS6$^{-/-}$: ratio = 0.8342, *P*-value = 0.1122; RPEp53$^{-/-}$ versus RPEp53$^{-/-}$STIL$^{-/-}$: ratio = 0.7798, *P*-value = 0.093], ARPC5L [RPEp53$^{-/-}$ versus RPEp53$^{-/-}$SAS6$^{-/-}$: ratio = 0.884, *P*-value = 0.2361; RPEp53$^{-/-}$ versus RPEp53$^{-/-}$STIL$^{-/-}$: ratio = 0.4299, *P*-value = 0.0091], ARPC3 [RPEp53$^{-/-}$ versus RPEp53$^{-/-}$SAS6$^{-/-}$: ratio = 1.1548, *P*-value = 0.0607; RPEp53$^{-/-}$ versus RPEp53$^{-/-}$STIL$^{-/-}$: ratio = 0.4517, *P*-value = 0.1221], ARPC4 [RPEp53$^{-/-}$ versus RPEp53$^{-/-}$SAS6$^{-/-}$: ratio = 1.1168, *P*-value = 0.2278; RPEp53$^{-/-}$ versus RPEp53$^{-/-}$STIL$^{-/-}$: ratio = 1.0888, *P*-value = 0.2754], and ARPC5 [RPEp53$^{-/-}$ versus RPEp53$^{-/-}$SAS6$^{-/-}$: ratio = 1.0969, *P*-value = 0.2859; RPEp53$^{-/-}$ versus RPEp53$^{-/-}$STIL$^{-/-}$: ratio = 0.6576, *P*-value = 0.0434]), profilin (Mouneimne et al, 2012) (RPEp53$^{-/-}$ versus RPEp53$^{-/-}$SAS6$^{-/-}$: ratio = 0.6461, *P*-value = 0.0183; RPEp53$^{-/-}$ versus RPEp53$^{-/-}$STIL$^{-/-}$: ratio = 0.2072, *P*-value = 0.0002), cofilin (Oser & Condeelis, 2009) (RPEp53$^{-/-}$ versus RPEp53$^{-/-}$SAS6$^{-/-}$: ratio = 0.6684, *P*-value = 0.068; RPEp53$^{-/-}$ versus RPEp53$^{-/-}$STIL$^{-/-}$: ratio = 0.2164, *P*-value = 0.00001), and the actin monomer–binding protein CAP1 (Bertling et al, 2004) (RPEp53$^{-/-}$ versus RPEp53$^{-/-}$SAS6$^{-/-}$: ratio = 0.3477, *P*-value = 0.0067; RPEp53$^{-/-}$ versus RPEp53$^{-/-}$STIL$^{-/-}$: ratio = 0.113, *P*-value = 0.0076). Immunoblotting of the FA fractions was able to validate the fact that there was negative regulation by centrosome assembly of Rac1, TRIO, GEF-H1, and integrin β1 in FAs, whereas disruption of the centrosome's intact structure resulted in a significant increase in these proteins in the FA fractions, in both cases compared with the

**Figure 3. The absence of the centrosome changes FA dynamics in a migrating cell.**
**(A)** RPEp53$^{-/-}$, RPEp53$^{-/-}$SAS6$^{-/-}$, and RPEp53$^{-/-}$STIL$^{-/-}$cells were plated on coverslips coated with fibronectin (10 µg/ml) for 15 min and then images were captured by phase-contrast microscopy. Scale bar, 20 µm. **(B)** The plot shows the area of cells spreading on coverslips coated with fibronectin (10 µg/ml) for 15 min. Data are mean ± SEM (n = 200 cells for each conditions). ***P < 0.001, compared with RPEp53$^{-/-}$. **(C)** RPEp53$^{-/-}$, RPEp53$^{-/-}$SAS6$^{-/-}$, and RPEp53$^{-/-}$STIL$^{-/-}$cells were plated on 96-well plates coated with fibronectin (10 µg/ml) for 30 min and then their cell attachment was measured. Fold of cells (relative to RPEp53$^{-/-}$ cells) that remained attached to the fibronectin-coated plates. Data are mean ± SEM (n = 6 independent experiments). ***P < 0.001, compared with RPEp53$^{-/-}$. **(D)** Size distribution of segmented FAs from RPEp53$^{-/-}$, RPEp53$^{-/-}$SAS6$^{-/-}$, and RPEp53$^{-/-}$STIL$^{-/-}$cells, before treatment with nocodazole (control) and at 0, 3, 10, and 60 min after nocodazole washout. Data are mean ± SEM (n = number of FAs/number of cells). **P < 0.01, compared with RPEp53$^{-/-}$. **(E)** Top: time-lapse TIRF microscopy images of GFP-paxillin during FA turnover in migrating RPEp53$^{-/-}$, RPEp53$^{-/-}$SAS6$^{-/-}$, or RPEp53$^{-/-}$STIL$^{-/-}$cells as indicated. Scale bar, 1 µm. Bottom: normalized fluorescent paxillin intensity over time in FA marked with arrows in the images. **(F)** FA lifetime in RPEp53$^{-/-}$, RPEp53$^{-/-}$SAS6$^{-/-}$, and RPEp53$^{-/-}$STIL$^{-/-}$cells as indicated (RPEp53$^{-/-}$: 54.84 ± 3.13 min; RPEp53$^{-/-}$SAS6$^{-/-}$: 44.29 ± 2.37 min; and RPEp53$^{-/-}$STIL$^{-/-}$: 36.25 ± 2.27 min). **(G)** The duration of FA assembly, maturation, and disassembly in RPEp53$^{-/-}$, RPEp53$^{-/-}$SAS6$^{-/-}$, and RPEp53$^{-/-}$STIL$^{-/-}$ cells as indicated. In (F) and (G), RPEp53$^{-/-}$: 46 FAs/7 cells; RPEp53$^{-/-}$SAS6$^{-/-}$: 44 FAs/7 cells; and RPEp53$^{-/-}$STIL$^{-/-}$: 46 FAs/7 cells, data are mean ± SEM. *P < 0.05; **P < 0.01; ***P < 0.001; NS, no significance, all compared with RPEp53$^{-/-}$. **(H)** TIRF microscopy images of immunolocalized α-tubulin (green), paxillin (to visualize FAs; red), and phalloidin (to visualize F-actin; blue) in RPEp53$^{-/-}$, RPEp53$^{-/-}$SAS6$^{-/-}$, and RPEp53$^{-/-}$STIL$^{-/-}$cells. Scale bar, 10 µm. The 8 × 8 µm areas indicated in the upper-left images are magnified in the merged and right three images.

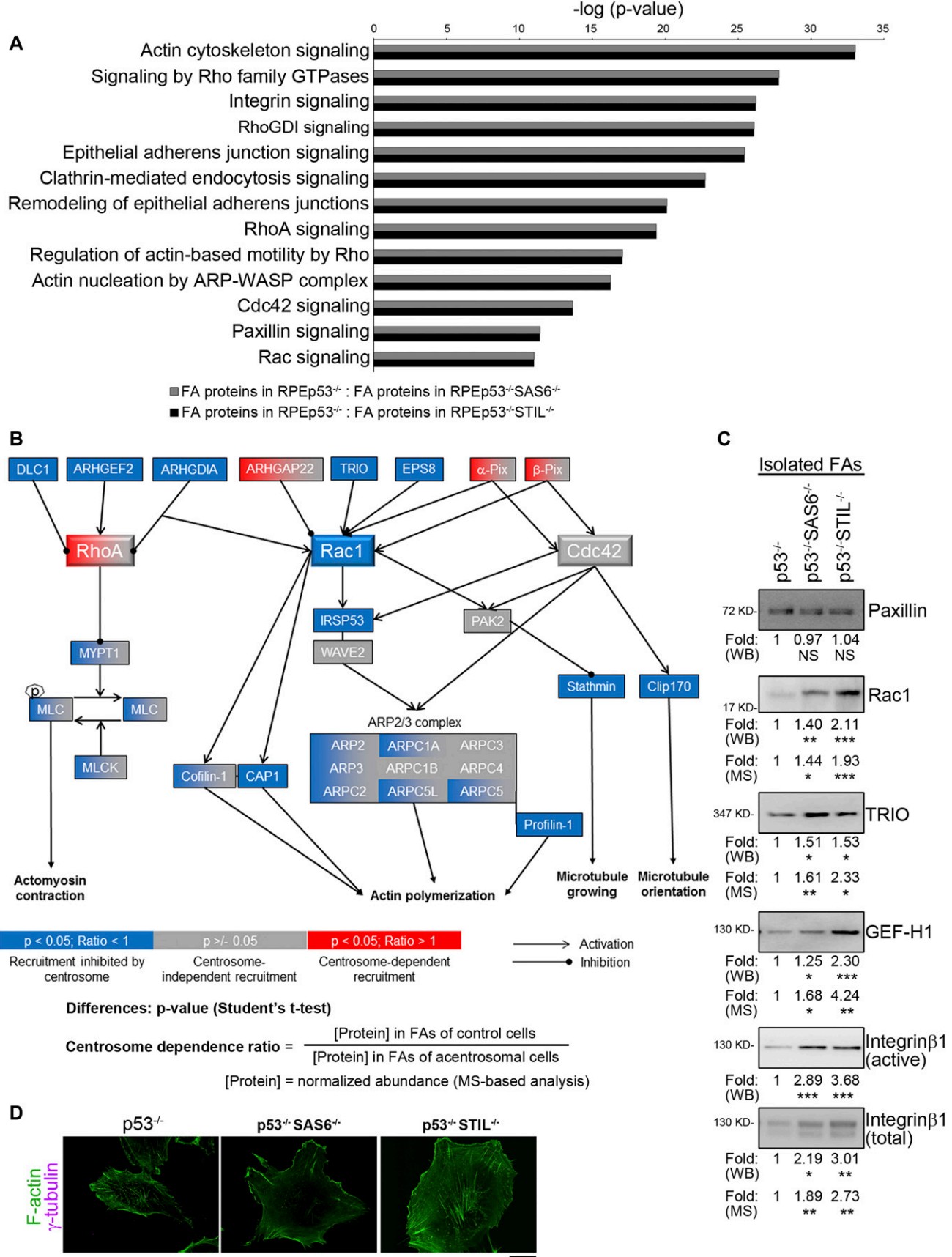

control cells (RPEp53$^{-/-}$). These results are similar to those obtained by the proteome analysis (Fold [MS] = 1/centrosome dependence ratio) (Fig 4C). Interestingly, active integrin $\beta$1 accumulation in FAs was also found to be regulated by the centrosome based on the Western blotting of isolated FAs, which revealed an increased accumulation of active integrin $\beta$1 in the FA fractions from acentrosomal cells (RPEp53$^{-/-}$SAS6$^{-/-}$ and RPEp53$^{-/-}$STIL$^{-/-}$) (Fig 4C) compared with control cells, supporting the results in Fig 3. These results were that centrosome disruption promotes FA formation, increase in the number of FAs, and increased cell adhesive ability. To further confirm if the centrosome modulates the downstream effect of Rac1, various cells were immunostained to detect $\gamma$-tubulin and F-actin, and the results showed that loss of the centrosome increased actin-based membrane protrusion (Fig 4D).

### Loss of the centrosome promotes excessive Rac1 activation and increased lamellipodial protrusion

To better understand how the centrosome may negatively regulate Rac1 signaling and lamellipodial protrusion, we focused on the effect of centrosome disruption on the modulation of Rac1 activation via an exploration of this protein's role in the membrane protrusion/retraction cycle. Rac1-GTP pull-down assays were used to show that Rac1 activity was increased in acentrosomal cells (RPEp53$^{-/-}$SAS6$^{-/-}$ and RPEp53$^{-/-}$STIL$^{-/-}$) compared with control cells (RPEp53$^{-/-}$) (Fig 5A). Next, we investigated the spatial distribution of active Rac1. To do this, control and acentrosomal cells were transfected with the Rac1 FRET biosensor pTriEx4-Rac1-2G (Fritz et al, 2015) and then imaged to assess the Rac1 FRET/ECFP ratio. The results revealed that centrosome disruption resulted in extensive Rac1 activation across multiple regions, whereas active Rac1 was limited to the front of the control cells (Fig 5B). We next studied cell membrane dynamics in the control and acentrosomal cells. To do this, we performed live-cell imaging of fluorescently labeled CAAX (YFP-CAAX), which labels cell membrane. In the time-lapse image series obtained using YFP-CAAX, we tracked the dynamics of the cell boundary using a previously described computational platform CellGeo (Tsygankov et al, 2014) to assess changes in the cell boundary. These changes were colored to show areas of protrusion (white) versus areas of retraction (black) (Fig 5C). In the control cells, there was a front/back asymmetric distribution of persistent protrusion/retraction, which indicates polarized cell morphology. On the other hand, the membrane dynamics of the

acentrosomal cells was random. Thus, the loss of the centrosome significantly enhanced Rac1 activation all over the cell, and this resulted in a large number of extensive lamellipodial protrusions, which suggests a possible link between Rac1 activation and microtubules in cells that have undergone centrosome disruption.

To investigate if microtubules contribute to the Rac1 activation induced by centrosome disruption, we carried out Rac1-GTP pull-down assays to examine Rac1 activity in the control and acentrosomal cells after treatment with nocodazole. The results revealed that Rac1 activity increased in acentrosomal cells, compared with control cells, and that the Rac1 activation induced by centrosome disruption was abolished by nocodazole treatment (Fig 5D). Thus, the presence of growing microtubules was a requirement for excessive Rac1 activation in acentrosomal cells, which implies that the increase in acentrosomal microtubules is delivering the specific signaling proteins that activate Rac1 excessively.

### The presence of TRIO on acentrosomal microtubules induces Rac1 activation and promotes lamellipodial protrusions

To understand how growing microtubules are able to regulate Rac1 activity in a migrating acentrosomal cell, we focused on the GEFs (TIAM1, $\beta$-PIX, and TRIO), the GAPs (GIT1 and GIT2), and the GDIs (ARHGDIA and GDI2) within the Rho-family small GTPases. Examining Rac1 activity revealed that in cells expressing control shRNAs (non-silencing), Rac1 activity was increased by loss of the centrosome, whereas Rac1 activation induced by centrosome disruption was abolished in cells having undergone TRIO shRNA knockdown (Fig 6A), not in cells expressing TIAM1, $\beta$-PIX, GIT1, GIT2, ARHGDIA, or GDI2 knockdown shRNAs (Table S6). Immunolocalization of F-actin and the lamellipodia marker cortactin in above cells revealed that loss of the centrosome resulted in extensive enhancement of lamellipodial protrusions (Figs 2G and 6B) and that this was abrogated in cells expressing shRNAs that inhibited TRIO expression (Fig 6B). In addition, wound-healing migration assays revealed that silencing of TRIO reversed the defect in directional persistence induced by centrosome disruption (Fig 6C). Thus, TRIO is required for excessive Rac1 activation and the increase in lamellipodial formation, induced by centrosome disruption.

Dynamic microtubules control TRIO's GEF activity (van Haren et al, 2014), so we hypothesized that TRIO may preferentially be associating with acentrosomal microtubules to determine the subcellular localization of its GEF activity. To test this, we isolated

---

**Figure 4.   The absence of the centrosome modulates actin filaments organization via FA signaling in a migrating cell.**
**(A)** The bar graph (scaled according to the ANOVA *P*-value) is representative of the results from the Ingenuity Pathway Analysis of the proteomics data from the various FA fractions obtained from RPEp53$^{-/-}$: RPEp53$^{-/-}$SAS6$^{-/-}$ (gray) or RPEp53$^{-/-}$: RPEp53$^{-/-}$STIL$^{-/-}$ cells (black); this shows the most dysregulated canonical pathways that are involved in migratory signaling. **(B)** Collective modulation by the centrosome of the abundance of small GTPases pathway proteins in the FAs. Proteins are represented by boxes that are color-coded according to their statistical significance categories and the magnitude of their ratio (as is indicated below each diagram). A *P*-value < 0.05 (*t* test) is considered to indicate with high confidence a significant change. Of the proteins with *P*-value < 0.05, the proteins with centrosome dependence ratio > 1 were considered to undergo centrosome-dependent recruitment, whereas the proteins with centrosome dependence ratio < 1 were considered to have their recruitment inhibited by the centrosome. **(C)** FA fractions from RPEp53$^{-/-}$, RPEp53$^{-/-}$SAS6$^{-/-}$, and RPEp53$^{-/-}$STIL$^{-/-}$ cells were analyzed by Western blotting using antibodies against paxillin, Rac1, TRIO, GEF-H1, active integrin $\beta$1, and total integrin $\beta$1. Fold (WB) (RPEp53$^{-/-}$: RPEp53$^{-/-}$SAS6$^{-/-}$: RPEp53$^{-/-}$STIL$^{-/-}$ cells) indicates the fold enrichment of paxillin (1: 0.97 ± 0.12: 1.04 ± 0.18; n = 12 independent experiments), Rac1 (1: 1.40 ± 0.13: 2.11 ± 0.25; n = 22 independent experiments), TRIO (1: 1.51 ± 0.24: 1.53 ± 0.25; n = 10 independent experiments), GEF-H1 (1: 1.25 ± 0.10: 2.30 ± 0.29; n = 16 independent experiments), active integrin $\beta$1 (1: 2.89 ± 0.44: 3.68 ± 0.40; n = 11 independent experiments), and total integrin $\beta$1 (1: 2.11 ± 0.47: 2.82 ± 0.50; n = 9 independent experiments) in isolated FA fractions as determined by Western blotting (loaded with equal amount of total protein, data are mean ± SEM). Fold (MS) indicates fold enrichment of the indicated proteins in the various isolated FA fractions as determined by LC-MS/MS. **(D)** Confocal images of immunolocalized $\gamma$-tubulin (purple) and phalloidin (green) in RPEp53$^{-/-}$, RPEp53$^{-/-}$SAS6$^{-/-}$, and RPEp53$^{-/-}$STIL$^{-/-}$ cells. Scale bars, 20 $\mu$m.

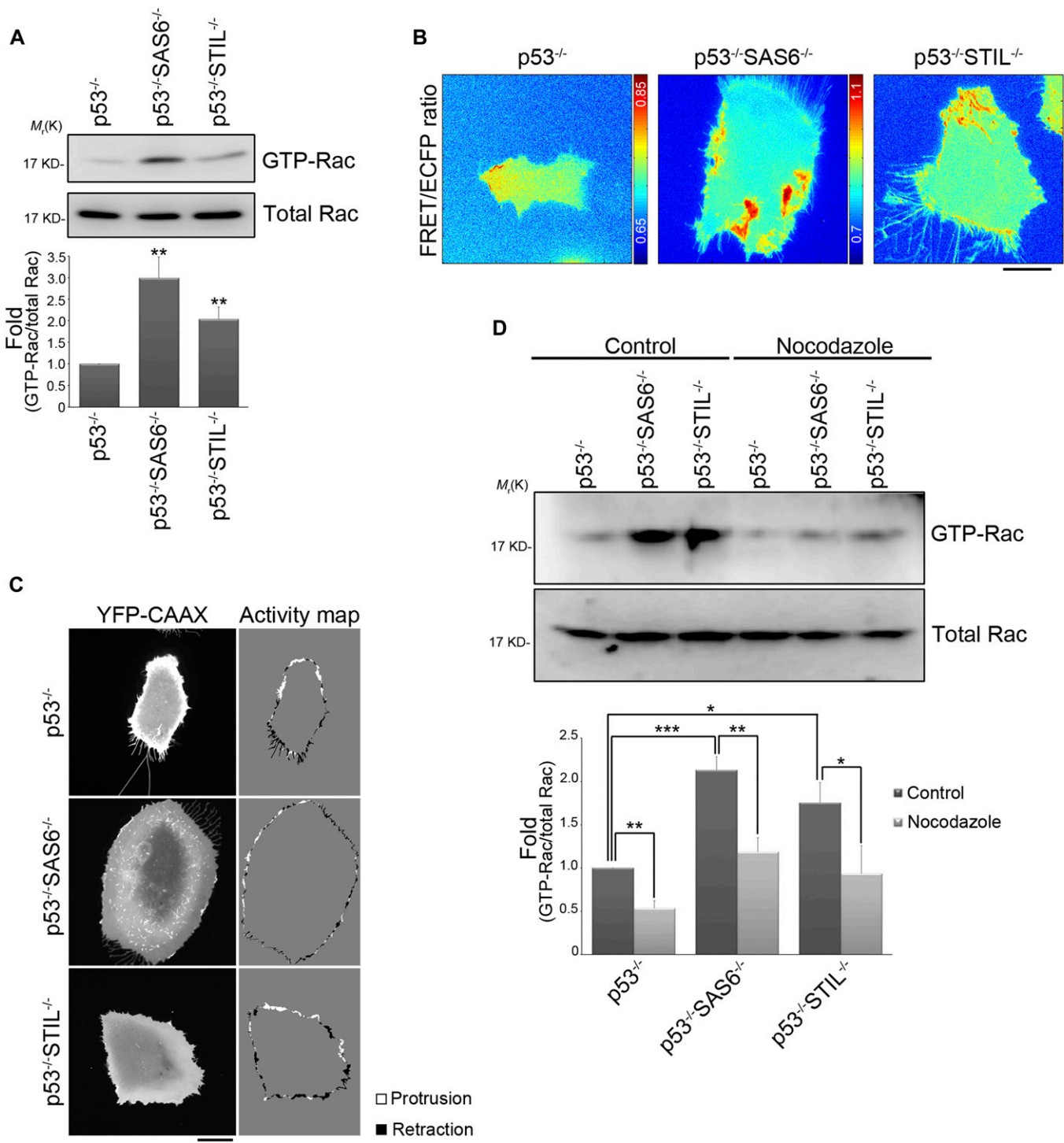

**Figure 5. The absence of the centrosome promotes small GTPase Rac1 activation via acentrosomal microtubules.**
**(A)** The effects of the centrosome on Rac1 activation. The level of GTP-bound Rac1 isolated from lysates of RPEp53$^{-/-}$, RPEp53$^{-/-}$SAS6$^{-/-}$, and RPEp53$^{-/-}$STIL$^{-/-}$cells by GST-PAK-CRIB pull-down and the amount Rac1 protein (total Rac) present in the input lysate were detected by Western blotting (top). Fold changes in the level of GTP-Rac relative to total Rac determined by Western blotting is indicated below the blots. Data are mean ± SEM (n = 8 independent experiments). \*\*$P$ < 0.01, compared with RPEp53$^{-/-}$. **(B)** Rac1 FRET/ECFP ratio images using the Rac1 biosensor (pTriEx4-Rac1-2G) in RPEp53$^{-/-}$, RPEp53$^{-/-}$SAS6$^{-/-}$, and RPEp53$^{-/-}$STIL$^{-/-}$cells. The cold and hot colors in the color bar represent low and high Rac1 activity levels, respectively. Bar, 20 μm. **(C)** Images of YFP-CAAX and a cell protrusion/retraction (white/black, respectively) dynamics map of migrating RPEp53$^{-/-}$, RPEp53$^{-/-}$SAS6$^{-/-}$, and RPEp53$^{-/-}$STIL$^{-/-}$cells. Bar, 20 μm. **(D)** Effects of growing microtubules on Rac1 activation. The level of GTP-bound Rac1 isolated from lysates of RPEp53$^{-/-}$, RPEp53$^{-/-}$SAS6$^{-/-}$, and RPEp53$^{-/-}$STIL$^{-/-}$cells treated with DMSO (control) or nocodazole (10 μM, 16 h) by GST-PAK-CRIB pull-down and the protein level of Rac1 (Total Rac) in the input lysate were detected by Western blotting (top). Fold changes in the levels of GTP-Rac relative to total Rac determined by Western blotting is indicated below the blots. Data are mean ± SEM (n = 5 independent experiments). \*$P$ < 0.05; \*\*$P$ < 0.01; \*\*\*$P$ < 0.001.

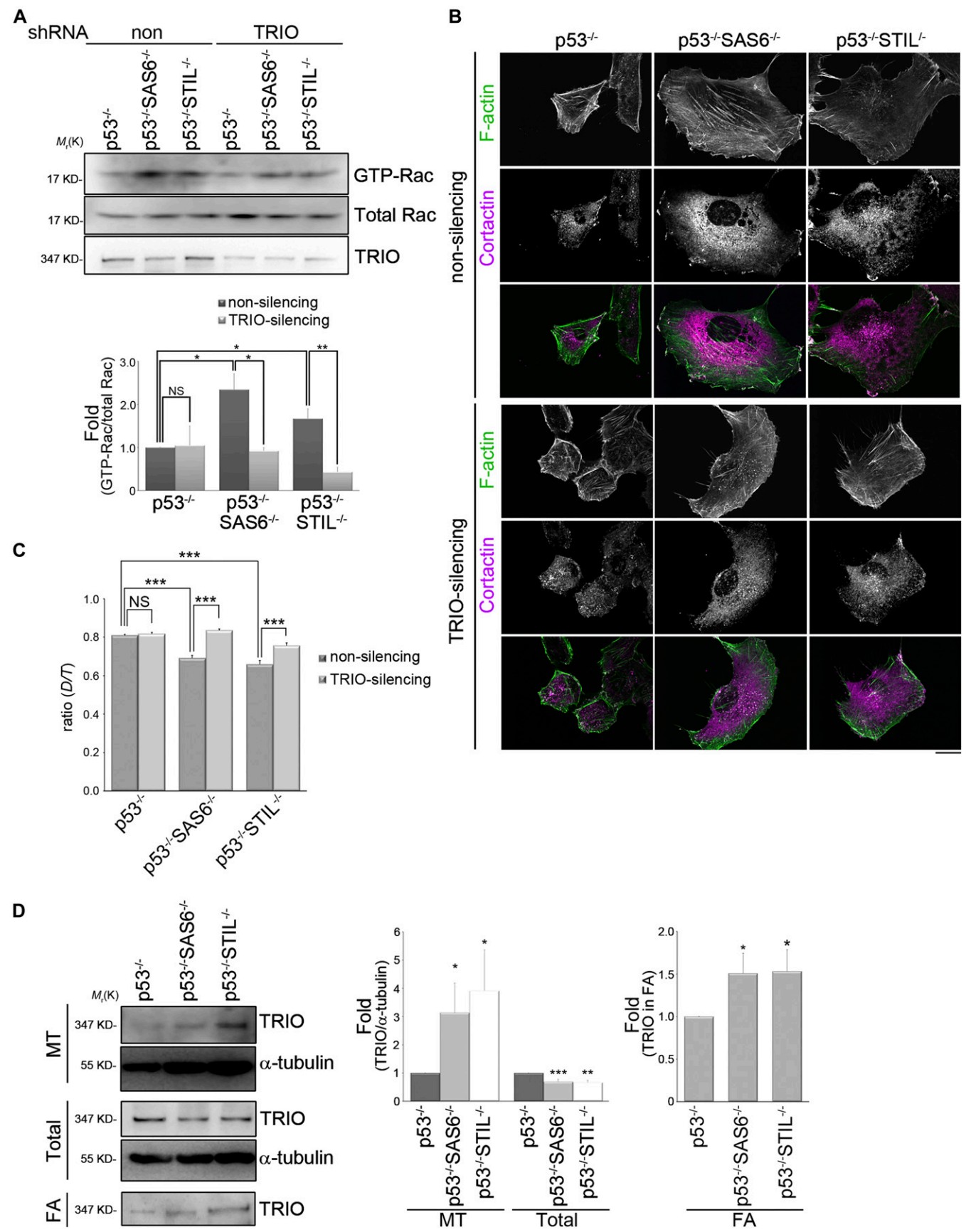

microtubule fractions from control (RPEp53$^{-/-}$ cells) and acentrosomal cells (RPEp53$^{-/-}$SAS6$^{-/-}$ and RPEp53$^{-/-}$STIL$^{-/-}$ cells). Immunoblotting of these microtubule fractions showed that there was increased TRIO accumulation in polymerized microtubules when there was centrosome disruption. Specifically, the loss of the centrosome resulted in a 3.1-fold increase in the presence of TRIO in the polymerized microtubules of RPEp53$^{-/-}$SAS6$^{-/-}$ cells and a 3.9-fold increase in the presence of TRIO in the polymerized microtubules of RPEp53$^{-/-}$STIL$^{-/-}$cells, both compared with control cells (RPEp53$^{-/-}$ cells) (Fig 6D). Next, we isolated the various FA fractions to confirm the results shown in Fig 4B and C, namely, that TRIO was enriched in the FAs from acentrosomal cells (Fig 6D). It was hypothesized by us that TRIO is delivered to FAs via the acentrosomal microtubules. To test this, we examined the FA localization of a TRIO mutant (TRIO-SRNN), which mutates the first SxIP motif (SRIP to SRNN) to abolish the binding of full-length TRIO to the microtubule plus end (van Haren et al, 2014). Analysis of mApple-paxillin and GFP-TRIO versus GFP-TRIO-SRNN revealed that GFP-TRIO, but not GFP-TRIO-SRNN, was localized in the paxillin-marked FAs of acentrosomal cells (RPEp53$^{-/-}$SAS6$^{-/-}$ cells) (Fig 7A). These findings indicate that the delivery of TRIO to FAs depends on its interaction with the plus ends of acentrosomal microtubules. To further determine whether acentrosomal microtubule-delivered TRIO is required for its GEF activity, we altered TRIO expression and analyzed the effect of this on directed cell migration. Wound-healing migration assays revealed that in cells expressing the control shRNA, the directional persistence of migrating cells was suppressed by centrosome disruption (Figs 1D, 6C, and 7B). On the other hand, these polarity defects were rescued by TRIO knockdown (Figs 6C and 7B). This recovery was suppressed by reexpressing GFP-TRIO, but not reexpression of GFP-TRIO-SRNN (Fig 7B). Taken together, these findings indicate that dynamic acentrosomal microtubules are able to deliver TRIO to the targeted FAs and this positively regulates TRIO's GEF activity, which in turn increases the activation of Rac1, enhances lamellipodia formation, and negatively regulates cell polarity during directed cell migration.

## Discussion

Our study profiles centrosome-dependent FA composition changes and has uncovered a new role for the centrosome in regulating balance within the microtubule system that allows correct FA signaling and the maintenance of directed cell migration. To understand the role of the centrosome in FA-mediated signaling, we profiled the centrosome-dependent FA proteome, and this revealed that the centrosome collectively modulates the abundance of various functional modules of proteins (Fig 4). We then focused on small GTPase Rho signaling and showed that the loss of the centrosome increases the abundance of proteins involved in the Rac1 regulatory module of the FAs, including the protein Rac1 GEF TRIO. Loss of the centrosome was found to promote the formation of acentrosomal microtubules (Fig 2B), which are involved in delivering TRIO to FAs (Fig 7A). In acentrosomal cells, microtubule-delivered TRIO governs specific aspects of FA signaling, enhances Rac1 activation, and disrupts directed cell migration. In the present study, we have demonstrated for the first time that the presence of a centrosome restricts the assembly of acentrosomal microtubules by forming centrosomal microtubules, which in turn leads to correct activation of Rac1 and appropriate directed cell migration (Fig 7C).

The discovery of the centrosome regulates cell polarization in a migrating cell via acentrosomal microtubule-controlled FA signaling is surprising. In spite of the fact that the centrosome has been known for a considerable time to control cell polarization and motility, it has been generally thought that the centrosome releases signals that affect polarization. Experiments using laser micro-irradiation in 1984 provides the first evidence for the importance of the centrioles in directed cell migration (Koonce et al, 1984). Several studies using advanced techniques to carry out centrosome ablation have also confirmed that one of the centrosome's functions is the maintenance of a polarized microtubule network that allows cell migration in a specific direction (Wakida et al, 2010; Zhang & Wang, 2017). Furthermore, recent evidence has suggested that the centrosome modulates motile cell polarization via emerging acentrosomal microtubules (Vinogradova et al, 2012). Our findings support this notion because we have found that centrosome loss promotes the assembly of acentrosomal microtubules that then attach to FAs; this up-regulates a Rac1 regulatory module in FAs and significantly enhances Rac1 activity. The result is that the cell loses its ability to maintain polarization during directed cell migration. Thus, our results provide the missing link that explains how the centrosome controls the coupling between Rac1 and actin polymerization at the cell front via a positive feedback loop to ensure cell polarity stability; specifically, this is done by restricting the assembly of acentrosomal microtubules. However, whether other mechanisms related to FA-mediated signaling are able to alter cell polarization or whether this is related to Cdc42 activity in acentrosomal cells has not yet been determined.

---

Figure 6.   The expression of TRIO is required for centrosome-modulated Rac1 activation.
**(A)** Effects of TRIO expression on centrosome-regulated Rac1 activation. RPEp53$^{-/-}$, RPEp53$^{-/-}$SAS6$^{-/-}$, and RPEp53$^{-/-}$STIL$^{-/-}$cells expressing non-silencing (non) or TRIO-silencing (TRIO) shRNAs were analyzed. The level of GTP-bound Rac1 isolated from lysates by GST-PAK-CRIB pull-down, the protein level of Rac1 (total Rac), and the amount of TRIO in the input lysate were detected by Western blotting (top). The fold changes in levels of GTP-Rac relative to total Rac determined by Western blotting are indicated below the blots. Data are mean ± SEM (n = 4 independent experiments). *P < 0.05; **P < 0.01; NS, no significance. **(B)** Confocal images of immunolocalized F-actin (green) and cortactin (to visualize lamellipodia; purple) in RPEp53$^{-/-}$, RPEp53$^{-/-}$SAS6$^{-/-}$, and RPEp53$^{-/-}$STIL$^{-/-}$cells expressing either non-silencing shRNA or TRIO-silencing shRNA. Scale bar, 20 $\mu$m. **(C)** Directional persistence of RPEp53$^{-/-}$, RPEp53$^{-/-}$SAS6$^{-/-}$, and RPEp53$^{-/-}$STIL$^{-/-}$cells expressing either non-silencing shRNA or TRIO-silencing shRNA. Data are mean ± SEM (non-silencing RPEp53$^{-/-}$, n = 100 cells; non-silencing RPEp53$^{-/-}$SAS6$^{-/-}$, n = 91 cells; non-silencing RPEp53$^{-/-}$STIL$^{-/-}$, n = 68 cells; TRIO-silencing RPEp53$^{-/-}$, n = 100 cells; TRIO-silencing RPEp53$^{-/-}$SAS6$^{-/-}$, n = 90 cells; TRIO-silencing RPEp53$^{-/-}$STIL$^{-/-}$, n = 81 cells). ***P < 0.001; NS, no significance. **(D)** (Left) Western blot analysis of the microtubule fractions (polymerized microtubule-containing fraction; MT), the total cell lysates (total), and the FA fractions (FA) isolated from RPEp53$^{-/-}$, RPEp53$^{-/-}$SAS6$^{-/-}$, and RPEp53$^{-/-}$STIL$^{-/-}$cells (loaded with equal total protein) using TRIO and $\alpha$-tubulin antibodies.; (middle) fold changes in the amount of TRIO:$\alpha$-tubulin in the isolated microtubule-containing fractions (MT) and in total cell lysates (total) were determined by Western blotting; (right) fold changes of TRIO in the FA fractions (FA) determined by Western blotting. Data are mean ± SEM (microtubule isolation, n = 17 independent experiments; total cell lysate, n = 6 independent experiments; FA isolation, n = 10 independent experiments). *P < 0.05, **P < 0.01, ***P < 0.001, compared with RPEp53$^{-/-}$.

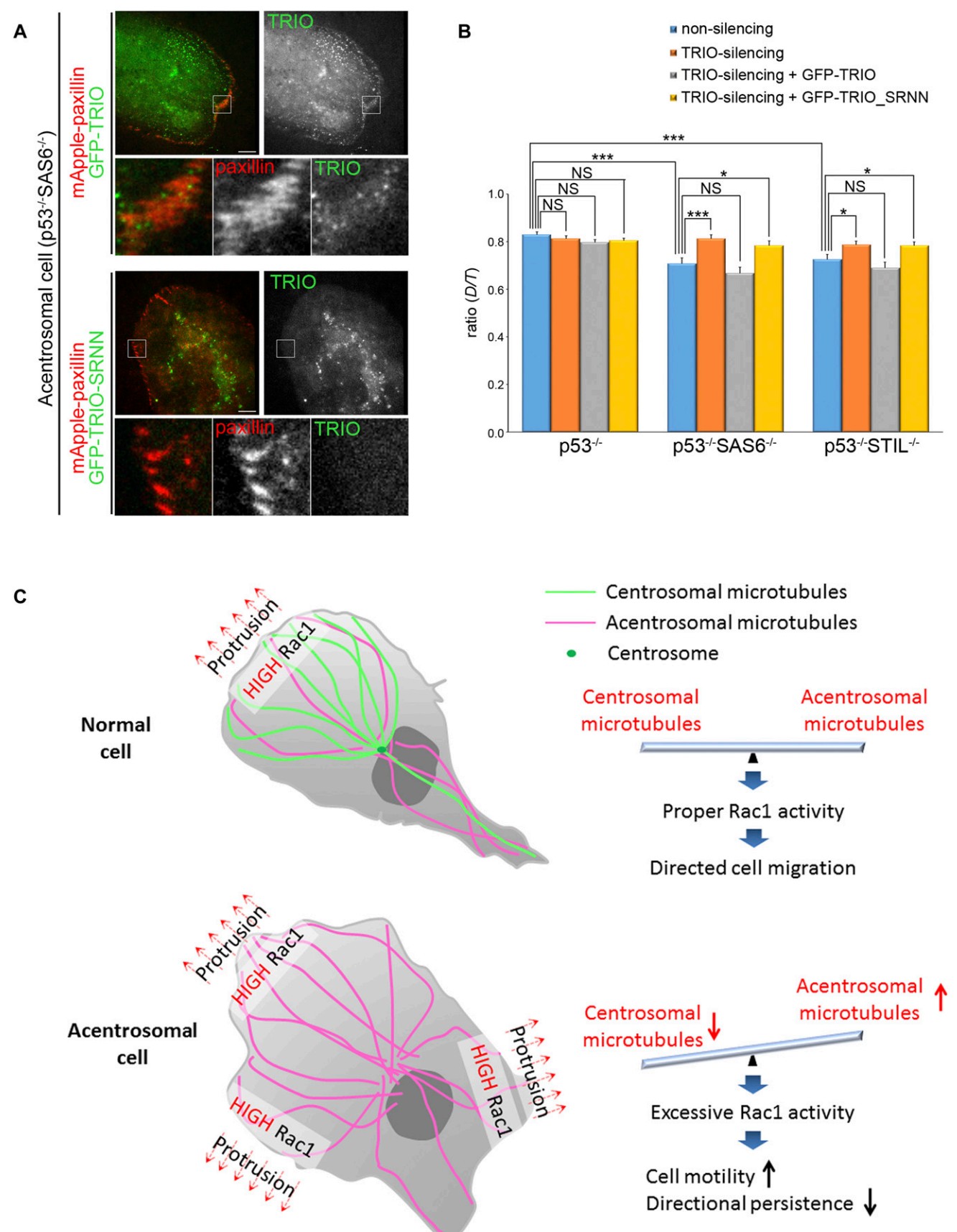

Our study has also revealed a previously unrecognized role for the Rac GEF TRIO protein; our findings place it in a physiologically relevant context related to centrosome-controlled cell polarity during directed cell migration. Compared with the control cells, TRIO shows a preference for associating with microtubules and FAs in acentrosomal cells (Fig 6D), which supports the notion that TRIO is likely to be a microtubule-associated protein involved in Rac1 activation (van Haren et al, 2014) and a component in FAs (Zaidel-Bar & Geiger, 2010; Zaidel-Bar et al, 2007) via its association with filamin A (Bellanger et al, 2000), FAK (Medley et al, 2003), LAR (Debant et al, 1996), Rac1 (Bateman et al, 2000; Bellanger et al, 1998; Debant et al, 1996; Gao et al, 2001; Newsome et al, 2000), and RhoA (Bellanger et al, 1998; Medley et al, 2000). We have confirmed that TRIO is delivered to FAs via its interaction with acentrosomal microtubules plus ends (Fig 7A), but we still do not know how TRIO is able to choose to target microtubules and FAs in acentrosomal cells. A recent study has revealed that in the absence of centrosome, a specific microtubule minus end–binding protein CAMSAP2 (calmodulin-regulated spectrin-associated protein) is involved in organizing and stabilizing microtubule minus ends to maintain the density and mass of acentrosomal microtubules (Wu et al, 2016). In addition, we were able to detect more acetylated tubulins within the polymerized microtubules of centrosome-depleted cells (Fig 2B). These findings indicate that acentrosomal microtubules most likely have specific posttranslational modifications and/or are able to associate with a number of specific proteins. Because posttranslational modifications that act through microtubule-associated proteins or motors are able to directly tune the stability and dynamics of microtubules (Portran et al, 2017), this may be why acentrosomal microtubules behave differently in terms of microtubule stability and dynamics compared with centrosomal microtubules. In addition, it is possible that the TRIO protein chooses to associate with microtubules and FAs in acentrosomal cells by recognizing specific associated proteins and/or by targeting the posttranslational modifications of microtubules and FAs in acentrosomal cells.

# Materials and Methods

### Cells

The human RPE p53$^{-/-}$, RPEp53$^{-/-}$SAS6$^{-/-}$, and RPEp53$^{-/-}$STIL$^{-/-}$ cells were kindly provided by Dr. Won-Jing Wang (National Yang-Ming University, Taiwan). RPEp53$^{-/-}$, RPEp53$^{-/-}$SAS6$^{-/-}$, and

RPEp53$^{-/-}$STIL$^{-/-}$ cells stably expressing non-silencing, TRIO, GIT1, GIT2, TIAM1, $\beta$-PIX, ARHGDIA, or GDI2 shRNA were generated using a lentiviral shRNA system according to the manufacturer's instructions (National RNAi Core Facility Platform/Academia Sinica; Open Biosystems/Thermo Fisher Scientific). A Nucleofector Kit V (Lonza) was used for transient transfections. For all experiments, the cells were seeded on 10 $\mu$g/ml fibronectin-coated coverslips or plates.

### Plasmids

Expression silencing of the following proteins was achieved using the following constructs: TRIO (TRCN0000196250; National RNAi Core Facility PlateForm), GIT1 (RHS4430-98911843; Open Biosystems), GIT2 (RHS4430-98476273; Open Biosystems), TIAM1 (RHS4430-98842140; Open Biosystems), $\beta$-PIX (RHS4430-98911853; Open Biosystems), ARHGDIA (TRCN0000008003; National RNAi Core Facility Platform), and GDI2 (RHS4430-99148519; Open Biosystems). The Rac1 biosensor (pTriEx4-Rac1-2G) was purchased from Addgene (#66110).

### Antibodies

Mouse anti-SAS6, rabbit anti-cortactin, and mouse anti-$\gamma$-tubulin: Santa Cruz; rabbit anti-STIL and mouse anti-TRIO: Abcam; rabbit anti-GEF-H1 and rabbit anti-integrin $\beta$1 (total form), mouse anti-GAPDH, rabbit anti-GOLPH2, rabbit anti-$\beta$-actin, rabbit anti-paxillin, mouse anti-$\gamma$-tubulin, and rabbit anti-NMIIA: GeneTex; mouse anti-$\alpha$-tubulin and mouse anti-acetylated tubulin: Sigma-Aldrich; rat anti-$\alpha$-tubulin: ABD; mouse anti-paxillin: BD; mouse anti-Rac1 and mouse anti-integrin $\beta$1 (active form): Millipore; rabbit anti-TRIO, Alexa Fluor 488 phalloidin, Alexa Fluor 488-anti-rat IgG, Alexa Fluor 488-anti-rabbit IgG, Alexa Fluor 488-anti-mouse IgG, Alexa Fluor 568-anti-mouse IgG, Alexa Fluor 568-anti-rabbit IgG, Alexa Fluor 680-anti-rabbit IgG, and DAPI: Thermo Fisher; and HRP-AffiniPure mouse anti-rabbit IgG and HRP-AffiniPure goat anti-mouse IgG: Jackson ImmunoResearch.

### Reagents

Nocodazole (Sigma-Aldrich).

### FA isolation

The protocol for FA isolation was carried out as described previously (Kuo et al, 2011, 2012). Briefly, RPE cells were plated on

---

**Figure 7. Effects of TRIO on cell migration.**
**(A)** TIRF microscopy images of RPEp53$^{-/-}$SAS6$^{-/-}$ cells expressing mApple-paxillin (red) with either GFP-TRIO (green) or GFP-TRIO-SRNN (green). Scale bar, 10 $\mu$m. The 9.6-$\mu$m × 9.6-$\mu$m areas indicated in the upper images are magnified in the bottom three images. **(B)** Directional persistence of RPEp53$^{-/-}$, RPEp53$^{-/-}$SAS6$^{-/-}$, and RPEp53$^{-/-}$STIL$^{-/-}$cells expressing non-silencing shRNA or TRIO-silencing shRNA alone or together with GFP-TRIO or GFP-TRIO-SRNN. Data are mean ± SEM (non-silencing RPEp53$^{-/-}$, n = 57 cells; non-silencing RPEp53$^{-/-}$SAS6$^{-/-}$, n = 55 cells; non-silencing RPEp53$^{-/-}$STIL$^{-/-}$, n = 51 cells; TRIO-silencing RPEp53$^{-/-}$, n = 61 cells; TRIO-silencing RPEp53$^{-/-}$SAS6$^{-/-}$, n = 49 cells; TRIO-silencing RPEp53$^{-/-}$STIL$^{-/-}$, n = 61 cells; TRIO-silencing RPEp53$^{-/-}$ cells expressing GFP-TRIO, n = 74 cells; TRIO-silencing RPEp53$^{-/-}$SAS6$^{-/-}$ cells expressing GFP-TRIO, n = 60 cells; TRIO-silencing RPEp53$^{-/-}$STIL$^{-/-}$ cells expressing GFP-TRIO, n = 54 cells; TRIO-silencing RPEp53$^{-/-}$ cells expressing GFP-TRIO-SRNN, n = 84 cells; TRIO-silencing RPEp53$^{-/-}$SAS6$^{-/-}$ cells expressing GFP-TRIO-SRNN, n = 60 cells; TRIO-silencing RPEp53$^{-/-}$STIL$^{-/-}$ cells expressing GFP-TRIO-SRNN, n = 53 cells). *$P$ < 0.05; ***$P$ < 0.001; NS, no significance. **(C)** Model showing the balance between centrosomal and acentrosomal microtubules during the control of Rac1 activation, FA dynamics, and directed cell migration. The centrosome acts as a controller and balances the formation of centrosomal and acentrosomal microtubules, which in turn restricts the random Rac1 activation caused by acentrosomal microtubules and activates Rac1 locally at cell front to induce membrane protrusion during directed cell migration. Interference with formation of the centrosome increases acentrosomal microtubules assembly, which results in the transportation of more TRIO protein, which in turn increases the excessive and random activation of Rac1. This then increases lamellipodia formation, leading to the loss of cell polarity and a negative impact on directed cell migration.

culture dishes coated with 10 μg/ml fibronectin for 24 h at 50% confluence. Hypotonic shock was induced by a 3-min treatment with TEA-containing low ionic strength buffer (2.5 mM triethanolamine [Sigma-Aldrich] in distilled water, pH 7.0). Cell bodies, including membrane-bound organelles, nuclei, cytoskeleton, and soluble cytoplasm, were removed by the use of a strong, pulsed hydrodynamic force using PBS containing protease inhibitors (Roche); this involved the use of a Waterpik (setting "3," Interplak dental water jet WJ6RW, Conair). Isolated FAs that remained bound to the dish were collected by scraping with a rubber policeman into 1× RIPA buffer (50 mM Tris–HCl, pH 8.0, 150 mM NaCl, 1.0% NP-40, 0.5% sodium deoxycholate, and 0.1% SDS), and the resulting solution was sonicated for 15 s on ice. Protein concentrations were measured by the Bradford protein assay (Bio-Rad).

## Protein identification for LC-MS/MS analysis

The precipitated FA protein pellets (~100 μg) were dissolved in resuspension buffer (6 M guanidine hydrochloride in 50 mM ammonium bicarbonate), mixed with dithiothreitol (final concentration 10 mM; Sigma-Aldrich) at 37°C for 3 h for reduction, and then mixed with iodoacetamide (final concentration 10 mM; Sigma-Aldrich) at room temperature for 1 h in dark for alkylation. Subsequently, the buffer in the FA protein mixture was replaced with 50 mM ammonium bicarbonate using a centrifugal filter device (Amicon Ultra-0.5, 10 kD; Millipore), then mixed with trypsin (Sequencing Grade; Promega) in a ratio of 1:50 (enzyme: protein mass ratio), and finally incubated at 37°C overnight. The trypsin digestion was terminated by adding formic acid (final concentration 0.5%, pH 2~3). Finally, the mixture of digested peptides was desalted using a ZipTip C18 (Millipore), dried, and stored at −80°C.

For LC-MS/MS analysis, the dried digested peptides were dissolved in 0.1% formic acid and analyzed using an UltiMate 3000 RSLC nanoflow LC system (Thermo Fisher Scientific) interfaced to an Orbitrap Fusion Tribrid Mass Spectrometer (Thermo Fisher Scientific) equipped with a PicoView nanosprayer (New Objective). The peptides were loaded directly onto a 25-cm × 75-μm C18 column (Acclaim PepMap RSLC; Thermo Fisher Scientific) and separated using a 40-min linear gradient of 100% mobile phase A (0.1% formic acid in water) to 35% mobile phase B (acetonitrile with 0.1% formic acid) at a flow rate of 500 nl/min. The mass spectrometer was operated in the data-dependent acquisition using a 3-s duty cycle. In detail, full-scan MS spectra (MS1) were acquired using the Orbitrap (m/z 350–1,600) with the resolution in 60,000 at m/z 400 and automatic gain control target at $2 \times 10^5$, followed by quadrupole isolation of precursors at 2 Th width for CID MS/MS fragmentation and detection in the linear ion trap (automatic gain control target at $1 \times 10^4$) with previously selected ions dynamically excluded for 60 s. Ions with single and an unrecognized charge state were also excluded. The LC-MS/MS raw files were normalized for label-free quantification by the software Progenesis QI for proteomics and searched using a Mascot Daemon 2.6.0 server. The mascot generic format (mgf) files were searched against the Swiss Prot (Swiss Institute of Bioinformatics) human database, carbamidomethyl (C) (variable), and oxidation (M) (variable). Up to two missed cleavages was allowed. The mass tolerance was set as ± 10 ppm for the MS

spectra and ± 0.6 D for the MS/MS spectra. For peptide identification, the false discovery rate was adjusted to 1% or less.

## Microtubule isolation

RPE cells were plated on culture dishes coated with 10 μg/ml fibronectin for 24 h at 50% confluence. The cells were then washed twice with PBS at 37°C and incubated with microtubule-stabilizing buffer (100 mM PIPES, pH 6.9, 5 mM $MgCl_2$, 2 mM EGTA, 2 M glycerol, 0.1% NP40, 10 mM β-glycerophosphate, 50 mM NaF, 0.3 μM okadaic acid, and 1 mM PMSF) containing protease inhibitors and phosphatase inhibitors (Roche) for 15 min at 37°C. Cell lysates were collected by scraping with a rubber policeman and the suspension centrifuged at room temperature for 5 min at 1,000 g. After centrifugation, the supernatants (soluble fraction) were collected and the pellets (insoluble fraction) were solubilized and sonicated in microtubule-stabilizing buffer for 15 s on ice.

## Rac1 activity assay

Cells were lysed in $Mg^{2+}$ lysis buffer (25 mM Hepes, pH 7.5, 150 mM NaCl, 1% Triton X-100, 10 mM $MgCl_2$, 1 mM EDTA, and 10% glycerol) containing protease inhibitors and phosphatase inhibitors (Roche). The cell lysates were then incubated with GST-PAK-CRIB coupled to glutathione-Sepharose 4 beads using RAC1 pull-down activation assay biochem kit (Cytoskeleton, Inc.) for 1.5 h at 4°C. The beads were washed in lysis buffer, re-suspended in SDS–PAGE sample buffer, and analyzed by Western blotting with anti-Rac1 antibody.

## Immunofluorescence analysis and image analysis

For α-tubulin/γ-tubulin/DAPI and F-actin/γ-tubulin staining, the cells were fixed with methanol at −20°C for 20 min and blocked with blocking solution (3% BSA/0.02% Triton X-100 in PBS) at room temperature for 1 h. Subsequently, the cells were incubated with the indicated primary antibodies in blocking solution at 4°C for 16 h and then incubated with fluorescent dye–conjugated secondary antibody at room temperature for 1 h. For α-tubulin/GOLPH2/DAPI staining, the cells were fixed with 4% paraformaldehyde at room temperature for 20 min, permeabilized with PBS containing 0.01% Triton X-100 and 0.05% SDS at room temperature for 5 min, and finally blocked with blocking solution (0.1% saponin and 0.2% BSA in PBS) at room temperature for 1 h. Subsequently, the cells were incubated with the indicated primary antibodies in blocking solution at 4°C for 16 h and then incubated with fluorescent dye–conjugated secondary antibody at room temperature for 1 h. For paxillin, α-tubulin/DAPI, α-tubulin/F-actin/DAPI, α-tubulin/F-actin/paxillin, and F-actin/cortactin staining, the cells were fixed with 4% paraformaldehyde in cytoskeleton buffer (10 mM MES pH 6.1, 138 mM KCl, 3 mM $MgCl_2$, and 2 mM EGTA) at room temperature for 20 min, permeabilized with cytoskeleton buffer containing 0.5% Triton X-100 at room temperature for 5 min, and blocked with blocking solution (3% BSA/0.02% Triton X-100 in PBS) at room temperature for 60 min. Subsequently, the cells were incubated with the indicated primary antibodies in blocking solution at 4°C for 16 h and then incubated with fluorescent dye–conjugated secondary antibody at room temperature for 1 h. Finally, the cells

mounted on a magnetic chamber (Live Cell Instrument) and incubated with PBS containing N-propyl gallate for confocal, TIRF, or epi-fluorescence imaging.

Confocal images were obtained using an *iLas* multi-modal of TIRF (Roper)/spinning disk confocal (CSUX1, Yokogawa) microscope (Ti-E; Nikon) system equipped with 40 × 1.3 NA, 60 × 1.49 NA, or 100 × 1.49 NA Plan objective lens (Nikon) and a Coolsnap HQ2 CCD (Photometrics). TIRF images were obtained using the same microscope system and either a 60 × 1.49 NA or 100 × 1.49 NA plan objective lens (Nikon) on an Evolve EMCCD (Photometrics) with an ~100-nm evanescent field depth. All confocal and TIRF images were captured and processed using Metamorph software. Epi-fluorescence images were obtained using a microscope (DMRBE; Leica) coupled with a 63 × NA 1.4 objective lens (Leica) and an 512B EMCCD (Andor) operated by Micro-Manager 1.4 software (Leica), or an epi-fluorescence microscope system (Ti-E; Nikon) coupled with a 60 × NA 1.49 plan objective lens (Nikon) and an sCMOS camera (OHCA-Flash 4.0, 1,024 × 1,024 pixels; Hamamatsu) operated by NIS-Elements software (Nikon).

To determine the FA area, TIRF images of paxillin-stained cells were thresholded to highlight only the FAs and the areas of the regions recorded using Metamorph. The area of recorded FAs was organized to give the adhesion size distribution. The results are presented graphically using Excel software (Microsoft).

### Time-lapse microscopy and image analysis

To analyze the dynamics of paxillin, cells expressing mApple-paxillin were mounted on a magnetic chamber (Live Cell Instrument) and incubated in phenol red–free culture medium with 25 mM Hepes (pH = 7.4) and imaged by TIRF using a 100 × 1.49 NA objective with an evanescent field depth of ~100 nm on the microscope system described above. Stage temperature was maintained at 37°C with an airstream incubator (Nevtek) and focus was maintained using the PerfectFocus system (Nikon). TIRF images of mApple-paxillin were captured at 1-min intervals using an Evolve EMCCD (Photometrics). To analyze the intensity changes of mApple-paxillin during FA turnover, the area of single FAs was hand-outlined in the paxillin channel over time. The integrated intensities within the areas and backgrounds around the areas were recorded for each time point. The background-subtracted, photobleach-corrected, and normalized (to the max value) intensity values were plotted as a function of time to determine when the intensity increases as the FA initiated and when the intensity decreases during termination. Thus, the duration of FA assembly, FA maturation, and FA disassembly was determined.

To analyze the dynamics of microtubules, cells expressing GFP-EB3 were mounted on a magnetic chamber (Live Cell Instrument) and incubated in phenol red–free culture medium with 25 mM Hepes (pH = 7.4) and imaged by spinning disk confocal using a 100 × 1.49 NA objective on the microscope system described above. Confocal images of GFP-EB3 were captured at 2-s intervals using an EMCCD (ProEM; Princeton). To track the dynamics of GFP-EB3, a microtubule plus end–tracking program (Applegate et al, 2011) was used to obtained microtubule growth speed and growth lifetime.

To image the recruitment of TRIO in FAs, cells expressing mApple-paxillin together with GFP-TRIO or GFP-TRIO-SRNN were

mounted on a magnetic chamber (Live Cell Instrument) and incubated in phenol red–free medium with 25 mM Hepes (pH = 7.4) and imaged by TIRF using a 100 × 1.49 NA objective with an evanescent field depth of ~100 nm on the microscope system that has been described above.

### FRET (Förster resonance energy transfer) analysis

Cells expressing FRET-based Rac1 biosensors (pTriEx4-Rac1-2G) were mounted on a magnetic chamber (Live Cell Instrument) and incubated in phenol red–free culture medium with 25 mM Hepes (pH = 7.4). The images were obtained by an epi-fluorescence microscope system (Ti-E; Nikon) equipped with a 40 × 1.30 NA objective lens (Nikon) and an ORCA-Flash4.0 V2 Digital CMOS camera (Hamamatsu) operated by NIS-Elements software (Nikon) with a 420DF20 excitation filter and two emission filters controlled by a filter changer (475DF40 for CFP and 535DF25 for YFP). The CFP images were collected using a 420DF20 excitation filter and a 475DF40 emission filter; the FRET images were collected using a 420DF20 excitation filter and a 535DF25 emission filter. Pixel-by-pixel analysis of the FRET/CFP ratio images was carried out based on background-subtracted fluorescence intensity images of the CFP and FRET images using Matlab (The MathWorks).

### Cell spreading assay and image analysis

Cells growing on tissue culture plates were trypsinized and re-seeded on coverslips coated with 10 $\mu$g/ml fibronectin for 30 min to allow them to adhere and spread. Next, the cells were fixed with 4% paraformaldehyde in PBS for 20 min at room temperature and then imaged using a microscope (Eclipse TS100; Nikon) coupled with a 20 × 0.45 NA objective lens (Nikon) and a WHITE CCD camera operated by ISCapture software (TUCSEN). To calculate the cell spreading area, the cell area was manually circled on the phase images using Metamorph image analysis software (Molecular Device) and the results are presented graphically using Excel software (Microsoft).

### Adhesion assay

The cell adhesion assays used 96-well plates that had been pre-treated with 1% denatured BSA at 37°C for 1 h and then coated with 10 $\mu$g/ml fibronectin. Cells growing on tissue culture plates were trypsinized, re-suspended in serum-free medium, and then re-seeded on the fibronectin-coated 96-well plates for 30 min or overnight (~16 h). After incubation, any non-attached cells were removed completely by washing with PBS twice and the adherent cells were fixed with 5% glutaraldehyde in $H_2O$ for 25 min at room temperature; this was followed by staining with 0.1% crystal violet in $H_2O$ for 25 min at room temperature. After removing any unbound crystal violet, the crystal violet–labeled adherent cells were solubilized in 50 $\mu$l solution A (50% ethanol and 0.1% acetic acid in $H_2O$) and the amount of crystal violet present measured using a Thermo Fisher Scientific Multiskan Spectrum at OD 550 nm. The results are presented graphically using Excel software (Microsoft).

### Wound-healing assay

Cells were growing on a chamber (with 500 $\mu$m space; SPL) embedded in 10 $\mu$g/ml fibronectin-coated 12-well plates in the culture medium for 16 h to create a wound in the confluent cells, and the plates were then placed in a temperature/$CO_2$-controlled chamber on a microscope (Axio Observer.Z1; Zeiss) equipped with a 10 × 0.25 NA objective lens (Zeiss). Time-lapse images were obtained at 15-min intervals over 12 h using a Rolera EM-C$^2$ EMCCD camera operated by Zen image analysis software (Zeiss). To calculate cell migration parameters, the centers of cell nuclei of cells at wound edge were manually tracked and positions recorded from the time-lapse image series using the "track points" function in Metamorph. Position and time data were then transferred to Excel to calculate the migration parameters, including cell migration speed and directional persistence. Speed was calculated as the total length of the migration path divided by the duration of migration. Directional persistence was calculated as the net migration distance divided by the total length of the migration path.

### Statistical analysis

Statistical significance was measured by either the $t$ test or the ANOVA test.

## Supplementary Information

## Acknowledgements

We thank the Academia Sinica Common Mass Spectrometry Facilities (Institute of Biological Chemistry, Academia Sinica, Taipei, Taiwan) for their help with the proteomics data analysis; Prof. Won-Jing Wang (National Yang-Ming University, Taipei, Taiwan) for providing the RPEp53$^{-/-}$, RPEp53$^{-/-}$SAS6$^{-/-}$, and RPEp53$^{-/-}$STIL$^{-/-}$cells; and Prof. Anne Debant (University of Montpellier, Montpellier, France) for providing the GFP-TRIO and GFP-TRIO-SRNN constructs. J-C Kuo is supported by research grants from the Taiwan Ministry of Science and Technology (MOST 103-2628-B-010-003-MY4; MOST 106-2633-B-010-002-; MOST 107-2633-B-010-001-; MOST 107-2320-B-010-049-), the Novel Bioengineering and Technological Approaches to Solve Two Major Health Problems in Taiwan sponsored by the Taiwan Ministry of Science and Technology Academic Excellence Program (MOST 107-2633-B-009-003), Cancer Progression Research Center (National Yang-Ming University) from The Featured Areas Research Center Program within the framework of the Higher Education Sprout Project by the Ministry of Education in Taiwan, the Yen Tjing Ling Medical Foundation, and the Ministry of Education's "Aim for the Top University Plan."

### Author Contributions

H-W Cheng: data curation, formal analysis, validation, and methodology.
C-T Hsiao: data curation, formal analysis, validation, and methodology.
Y-Q Chen: data curation, formal analysis, and methodology.
C-M Huang: data curation and formal analysis.
S-I Chan: data curation and formal analysis.
A Chiou: methodology and writing—review and editing.
J-C Kuo: conceptualization, data curation, formal analysis, supervision, funding acquisition, validation, investigation, project administration, writing—original draft, review, and editing.

### Conflict of Interest Statement

The authors declare that they have no conflict of interest.

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
