## [Reviewer comments · Life Science Alliance]

Centrosome guides spatial activation of Rac to control cell polarization and directed cell migration

Hung-Wei Cheng, Cheng-Te Hsiao, Yin-Quan Chen, Chi-Ming Huang, Seng-I Chan, Anne Debant5
Niels Galjart, Arthur Chiou, and Jean-Cheng Kuo
DOI: 10.26508/lsa.201800135

Review timeline:

Submission Date:	19 July 2018
Editorial Decision:	9 August 2018
Revision Received:	8 January 2019
Editorial Decision:	9 January 2019
Revision Received:	19 January 2019
Accepted:	28 January 2019

Report:

(Note: Letters and reports are not edited. The original formatting of letters and referee reports may not be reflected in this compilation.)

No Peer Review Process File is available with this article, as the authors have chosen not to make the review process public in this case.

1st Editorial Decision

9 August 2018

Thank you for submitting your manuscript entitled "Centrosome guides spatial activation of Rac to control cell polarization and directed cell migration" to Life Science Alliance. The manuscript was assessed by expert reviewers, whose comments are appended to this letter.

As you will see, while the referees think that your study offers an important examination of the role of the centrosome in coordinating cell migration, numerous points were also raised that would need to be addressed before the manuscript could be considered for publication. We think that the concerns raised are all valid and can get addressed in the normal revision timeframe by adding new data/better data for some of your figures, by adding controls, and by clarifying some aspects in the text. We would therefore like to invite you to submit a revised version of your manuscript. Importantly, we would expect for publication in Life Science Alliance that such a revised version better links TRIO to the observed effects (following the reviewers' constructive suggestions).

Thank you for this interesting contribution to Life Science Alliance. We are looking forward to receiving your revised manuscript.

2nd Editorial Decision

9 January 2019

Thank you for submitting your revised manuscript entitled "Centrosomes guide spatial activation of Rac to control cell polarization and directed cell migration". As you will see, reviewer #2 appreciates the introduced changes and now supports publication, and we would thus be happy to publish your paper in Life Science Alliance pending final revisions necessary to address this reviewer's comment on text changes needed and to meet our formatting guidelines:

- please carefully revise your manuscript text, I also attach a modified abstract below for guidance
- please provide the supplementary table file as a word docx
- please add legends for the three movies to your manuscript file
- please link your ORCID iD to your profile in our submission system, you should have received a message with instructions on how to do so

REVISED ABSTRACT:

Directed cell migration requires centrosome-mediated cell polarization and dynamical control of focal adhesions (FAs). To examine how FAs cooperate with centrosomes for directed cell migration, we used centrosome-deficient cells and found that loss of centrosomes enhanced the formation of acentrosomal microtubules, which failed to form polarized structures in wound-edge cells. In acentrosomal cells, we detected higher levels of Rac1-GEF TRIO on microtubules and FAs. Acentrosomal microtubules deliver TRIO to FAs for Rac1 regulation. Indeed, centrosome disruption induced excessive Rac1 activation around the cell periphery via TRIO, causing rapid FA turnover, a disorganized actin meshwork, randomly protruding lamellipodia as well as loss of cell polarity. This study reveals the importance of centrosomes to balance the assembly of centrosomal and acentrosomal microtubules and to deliver microtubule-associated TRIO proteins to FAs at the cell front for proper spatial activation of Rac1, FA turnover, lamellipodial protrusion and cell polarization, thereby allowing for directed cell migration.

3rd Editorial Decision

28 January 2019

Thank you for submitting your Research Article entitled "Centrosome guides spatial activation of Rac to control cell polarization and directed cell migration". It is a pleasure to let you know that your manuscript is now accepted for publication in Life Science Alliance. Congratulations on this interesting work.